# THE ADEMAMIX OPTIMIZER: BETTER, FASTER, OLDER

**Matteo Pagliardini** [†]
EPFL

**Pierre Ablin**
Apple

**David Grangier**
Apple

## ABSTRACT

Momentum based optimizers are central to a wide range of machine learning applications. These typically rely on an Exponential Moving Average (EMA) of gradients, which decays exponentially the present contribution of older gradients. This accounts for gradients being local linear approximations which lose their relevance as the iterate moves along the loss landscape. This work questions the use of a single EMA to accumulate past gradients and empirically demonstrates how this choice can be sub-optimal: a single EMA cannot simultaneously give a high weight to the immediate past, and a non-negligible weight to older gradients. Building on this observation, we propose AdEMAMix, a simple modification of the Adam optimizer with a mixture of two EMAs to better take advantage of past gradients. Our experiments on language modeling and image classification show— quite surprisingly—that gradients can stay relevant for tens of thousands of steps. They help to converge faster, and often to lower minima: e.g., a 1.3B parameter AdEMAMix LLM trained on 101B tokens performs comparably to an AdamW model trained on 197B tokens ($+95\%$). Moreover, our method significantly slows-down model forgetting during training. Our work motivates further exploration of different types of functions to leverage past gradients, beyond EMAs.

## 1 INTRODUCTION

With large neural networks, deep-learning has revolutionized numerous fields, such as computer vision and natural language processing. At the heart of this paradigm lies the challenge of optimizing complex, non-convex loss functions using noisy gradient estimates. This optimization process is typically carried out using variants of Stochastic Gradient Descent (SGD) (Robbins & Monro, 1951) or adaptive methods such as Adam and AdamW (Kingma & Ba, 2015; Loshchilov & Hutter, 2019), which have become ubiquitous in training state-of-the-art models (Devlin et al., 2019; Brown et al., 2020; Dosovitskiy et al., 2021a; Radford et al., 2021; Zhai et al., 2022; Dehghani et al., 2023; Touvron et al., 2023; Dubey et al., 2024).

A key component in many of these iterative optimization algorithms is momentum, which has long been shown to accelerate convergence (Nemirovskii & Nesterov, 1985) and often leads to solutions with superior generalization properties (Sutskever et al., 2013). By accumulating gradient vectors over successive optimization steps, momentum helps overcome small local variations of the loss landscape, potentially escaping shallow local minima, and accelerate in plateau regions (Qian, 1999; Ruder, 2016; Goh, 2017). Both SGD with momentum (SGD+M) and Adam incorporate momentum under the form of Exponential Moving Averages (EMAs) of past gradients $\mathcal{G}^T = \{\boldsymbol{g}^{(0)}, \ldots, \boldsymbol{g}^{(T)}\}$:

$$\text{EMA}(\beta, \mathcal{G}^T) \triangleq \beta \cdot \text{EMA}(\beta, \mathcal{G}^{(T-1)}) + (1-\beta)\boldsymbol{g}^{(T)} = \sum_{i=0}^{T} \beta^i (1-\beta)\boldsymbol{g}^{(T-i)}. \quad \text{(EMA)}$$

Two considerations support the use of EMAs. From a practical standpoint, the recursive formula of EMA allows for efficient implementations, which do not require maintaining a buffer of past gradients. From a theoretical standpoint, gradient descent with momentum leads to optimal convergence rates for quadratics (Polyak, 1964; Nesterov, 1983). However, those results do not guarantee any optimality for general non-quadratic cases (Goujaud et al., 2023).

---

[†]Work done while interning at Apple.

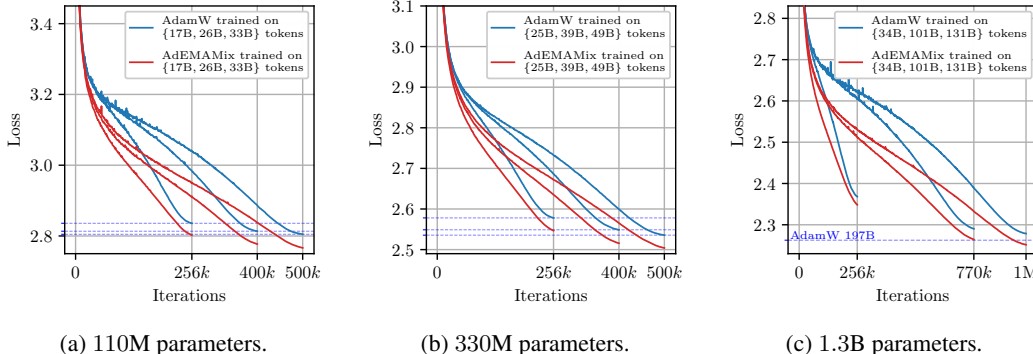

(a) 110M parameters.    (b) 330M parameters.    (c) 1.3B parameters.

Figure 1: **Comparing AdamW and AdEMAMix on language modeling.** In **(a,b,c)**, we plot the loss obtained using AdamW and AdEMAMix (our optimizer) to train Transformer models of various sizes on the Redpajama dataset. In **(a)**, we train multiple baselines for $256k$, $400k$, and $500k$ iterations, resulting in processing from 17B to 33B tokens. Two AdamW runs with different number of iterations look very different as we use a cosine-decay for the learning rate. We compare those baselines to training AdEMAMix for $256k$ iterations. We observe that our method reaches a similar loss as an AdamW model trained on nearly twice the number of tokens. Analogous comparisons can be derived from **(b)** and **(c)**. Notably, in **(c)**, a 1.3B parameter AdEMAMix model trained on 101B tokens performs comparably to an AdamW model trained on 197B tokens (95% more, blue horizontal line). See § 4.1 and App. B.1 for a detailed description of our experimental setup, including hyperparameters.

The use of momentum in optimization is grounded in the varying nature of gradients. As local linear approximations of the loss landscape, their information can quickly become outdated as the optimization process progresses (Pascanu et al., 2013). For this reason, practitioners typically employ moderate momentum values (i.e. $\beta \approx 0.8$ or $0.9$), effectively creating a moving average of recent gradients while discarding older information. Selecting larger $\beta$ values seems counter-intuitive, as it would suggest that older gradients maintain their relevance over extended periods of training. While it is tempting to see the use of small $\beta$s as a confirmation of the limited temporal relevance of gradients, our work reveals instead that older gradients can efficiently be used. When we increase $\beta$, we decrease the relative importance of recent gradients, and the iterate now fails to respond to local changes in the loss landscape. We observe that a single EMA cannot both give a significant weight to recent gradients, and give a non-negligible weight to older gradients (see Fig. 3a). However, a linear combination between a "fast-changing" (e.g. $\beta = 0.9$ or $\beta = 0$) and a "slow-changing" (e.g. $\beta = 0.9999$) EMA allows the iterate to beneficiate from (i) the great speedup provided by the larger (slow-changing) momentum, while (ii) still being reactive to small changes in the loss landscape (fast-changing). More precisely, we find the following statement to convey an important intuition behind this approach:

> *While changing the direction of the slow momentum is difficult, any adjustment orthogonal to that direction is easy—which favors fast progress in sinuous valley-like landscapes: the fast EMA helping to stay in the valley while the slow EMA accelerates in the valley's direction.*

A toy illustration of this can be seen in Fig. 2. Based on this idea, we propose AdEMAMix (**Ad**aptive **EMA Mix**ture), a novel Adam based optimizer which successfully leverages very old gradients to reach better solutions.

**Contributions.** Our contributions can be summarized as follows:

- We propose AdEMAMix, a novel optimizer which better leverages past gradients by avoiding a common pitfall of EMA-based optimizers (see § 3).

- We empirically demonstrate the superiority of our method over Adam by training ViT and language models (Transformers and Mamba) of up to 1.3B parameters (see § 4). In addition, we show gains from switching mid-training from Adam to AdEMAMix (see Fig. 5).

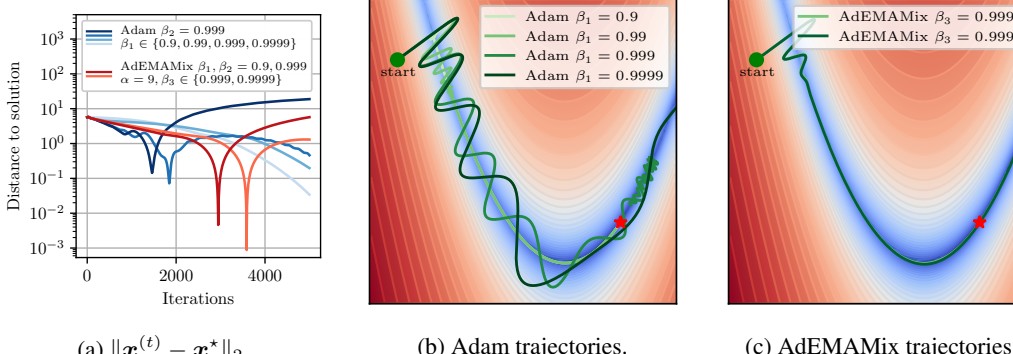

(a) $\|\boldsymbol{x}^{(t)} - \boldsymbol{x}^{\star}\|_2$.       (b) Adam trajectories.      (c) AdEMAMix trajectories.

Figure 2: **Comparing Adam and AdEMAMix on the Rosenbrock function.** Starting from $\boldsymbol{x}^{(0)} = [-3, 5]$, we minimize the Rosenbrock function $f(x_1, x_2) = (1 - x_1)^2 + 100(x_2 - x_1^2)^2$. The global minimum ($\star$) is $\boldsymbol{x}^{\star} = [1, 1]$. We use $\beta_2 = 0.999$ for Adam and $(\beta_1, \beta_2, \alpha) = (0.9, 0.999, 9)$ for AdEMAMix (see § 3). We reduce the learning rate for AdEMAMix to compensate for the influence of $\alpha$. We do a sweep over $\beta_1$ (resp. $\beta_3$) for Adam (resp. for AdEMAMix). In **(b)**, When Adam's $\beta_1$ is small (e.g. 0.9), the iterates do not oscillate but convergence is slow. Increasing $\beta_1$ makes the iterates move faster but with large oscillations. In contrast, for AdEMAMix in **(c)**, we observe that despite $\beta_3$ being large, the iterates moves fast and without oscillations. This results in reaching better solutions faster as can be seen in **(a)**.

- We show AdEMAMix forgets the training data slower when compared to Adam (see Fig. 4).

- More broadly, our findings contribute to a deeper understanding of the optimal balance between using historical gradients and adapting to the rapidly changing loss landscape. Our work invites further research in methods combining old and recent gradients, beyond EMAs.

## 2 RELATED WORK

**Work on understanding momentum.** From the seminal work of Polyak (1964), many publications analyzed the effect of gradient descent + momentum (GD+M) in both convex and non-convex settings (Ghadimi et al., 2015; Flammarion & Bach, 2015; Kidambi et al., 2018; Defazio, 2020; Sebbouh et al., 2021). While the acceleration in the noise-free setting has been long theorized for convex functions, several publications indicate this effect might not necessarily extend to stochastic settings (Yuan et al., 2016; Kidambi et al., 2018; Leclerc & Madry, 2020), emphasizing instead a link between momentum and effective learning rate. Recent work have been seeking to understand the impact of momentum on generalization through studying the implicit bias of momentum methods (Ghosh et al., 2023; Papazov et al., 2024), exposing a preference of SGD+M for lower norm solutions. Those further exposed a link between higher momentum and higher effective learning rate and higher variance reduction. Despite the volume of prior work on the subject, our understanding of momentum methods in non-convex stochastic settings is still incomplete (Yuan et al., 2016). Oscillatory behaviours, and the sometimes ambiguous effect of variance on optimization render the analysis tedious. From a theoretical standpoint, our work raises several questions. First, given that we gain from averaging very old gradients, what can it reveal of the loss landscape and the consistency of one batch's gradient during training? Second, would our approach not decrease the variance up to a point that is harming generalization (Ghosh et al., 2023)? While no answer to those questions is given in this work, we provide a toy justification which indicates that large momentums can have a positive impact in noise-free non-convex settings (see Fig. 2)—indicating the improvement of our approach is at least partially explainable without considering variance-reduction effects. We moreover expose a link between momentum and forgetting the training data (see Fig. 4), which to our knowledge is novel.

**Work on deep-learning optimizers.** Despite the popularity of Adam and AdamW (Kingma & Ba, 2015; Loshchilov & Hutter, 2019) in training deep neural networks, optimizer design is a rich field of research and we focus on a few of the works most relevant to this study. Adafactor (Shazeer & Stern, 2018) improves Adam's memory efficiency by factorizing the second moment estimate. Lamb (You

et al., 2020) extends Adam by adding layerwise normalized updates. Chen et al. (2023a) use algorithm discovery to derive the Lion optimizer. Contrary to Adam, Lion uses a single momentum term and the sign function to produce updates with the same magnitude across dimensions. Interestingly, Chen et al. (2023a) also report better scores are obtained when using a slightly larger momentum term ($\beta = 0.99$). In this work we show how increasing the momentum well beyond this value can still be beneficial. See App. C.3.2 for a detailed comparison between AdEMAMix and Lion. Recently, Liu et al. (2023) introduced Sophia, a scalable second-order optimizer designed for LLM training. Sophia uses a Hessian-based pre-conditioner which better normalizes the step size, penalizing steps in high curvature direction and accelerating in low curvature directions. Understanding in which circumstances those novel optimizers bring improvements is still being investigated (Kaddour et al., 2023), and Adam's dominance remains mostly unchallenged.

**Work incorporating an additional momentum term.** Lee et al. (2024) introduce Grokfast, which uses a pre-filtering step on the gradient to amplify the low frequencies and help solve groking. When combined with Adam, it effectively applies the Adam's EMAs on top of another gradient averaging method (e.g. EMA for Grokfast-EMA). Somewhat similarly, Chen et al. (2023b) refer to the Double EMA (DEMA) used in some finance applications (Mulloy, 1994) as one motivation for their AdMeta optimizer. Our motivation behind AdEMAMix is to combine both a high sensitivity to the recent gradients as well as incorporating very distant gradient, in this respect, using nested EMAs is not the right candidate as it reduces the influence of recent gradients. A more detailed review of AdMeta and DEMA is in App. C.3.1. Most relevant to us, Lucas et al. (2019, AggMo) also observe that using a combination of EMAs can enable the use of larger $\beta$s, and incorporate a sum of $K$ momentum terms into *GD*. They show their approach reaches similar performances as baseline optimizers, with a faster convergence. In contrast, we modify *Adam*, and introduce schedulers that are critical to reaching good performances at larger scales. This allows us to outperform Adam by a significant margin. In App. C.3.3, we notice no further improvement by adding more momentum terms. Finally, Szegedy et al. (2024) propose a general framework to derive and study optimizers with linear combinations of memory vectors—which encompasses EMA mixtures.

**Work on distributed optimization.** Perhaps surprisingly, recent work on distributed optimization—DiLoCo and SlowMo (Douillard et al., 2023; Wang et al., 2020)—are relevant to our work. $N$ workers $\boldsymbol{\theta}_1^{(t)}, \ldots, \boldsymbol{\theta}_N^{(t)}$ are trained independently for $K$ steps (e.g. $K = 500$). Every $K$ steps, the delta updates $\{\Delta\boldsymbol{\theta}_i\}_{i=1}^N \triangleq \{\boldsymbol{\theta}_i^{(t+K)} - \boldsymbol{\theta}_i^{(t)}\}_{i=1}^N$ are averaged and applied to each worker using an outer optimizer *with momentum* $\beta$: $\boldsymbol{\theta}_i^{(t+K)} = \boldsymbol{\theta}_i^{(t)} - \eta \cdot \text{Opt}(\frac{1}{N}\sum_i \Delta\boldsymbol{\theta}_i, \beta)$. The application of the outer momentum every 500 steps increases the importance of old gradients in the optimization trajectory. We believe this observation might in parts explain the strong results provided by those methods, and further motivated our work.

## 3 OUR METHOD: ADEMAMIX

**Setup & notations.** Let $\mathcal{L}_{\boldsymbol{\theta}} : \mathcal{X} \mapsto \mathbb{R}$ be a loss function parameterized by $\boldsymbol{\theta}$, and mapping inputs $\boldsymbol{x} \in \mathcal{X}$ to $\mathbb{R}$. Given a sampled batch $\boldsymbol{x}$, let $\boldsymbol{g} = \nabla_{\boldsymbol{\theta}}\mathcal{L}_{\boldsymbol{\theta}}(\boldsymbol{x})$ be a stochastic gradient of the loss w.r.t. $\boldsymbol{\theta}$. To minimize the empirical loss, the Adam optimizer (Kingma & Ba, 2015) relies on first and second moments, resp. $\boldsymbol{m}$ and $\boldsymbol{\nu}$, estimated via two EMAs parametrized by $(\beta_1, \beta_2) \in [0, 1[^2$. A weight-decay parameter $\lambda \in \mathbb{R}^+$ is often used as in Loshchilov & Hutter (2019):

$$\begin{cases} \boldsymbol{m}^{(t)} = \beta_1 \boldsymbol{m}^{(t-1)} + (1-\beta_1)\boldsymbol{g}^{(t)}, & \hat{\boldsymbol{m}}^{(t)} = \frac{\boldsymbol{m}^{(t)}}{1-\beta_1^t} \\ \boldsymbol{\nu}^{(t)} = \beta_2 \boldsymbol{\nu}^{(t-1)} + (1-\beta_2)\boldsymbol{g}^{(t)^2}, & \hat{\boldsymbol{\nu}}^{(t)} = \frac{\boldsymbol{\nu}^{(t)}}{1-\beta_2^t} \\ \boldsymbol{\theta}^{(t)} = \boldsymbol{\theta}^{(t-1)} - \eta\big(\frac{\hat{\boldsymbol{m}}^{(t)}}{\sqrt{\hat{\boldsymbol{\nu}}^{(t)}}+\epsilon} + \lambda\boldsymbol{\theta}^{(t-1)}\big). \end{cases} \quad \text{(AdamW)}$$

With $t > 0$ being the timestep, $\eta$ being the learning rate, and $\epsilon$ a small constant. Initially $\boldsymbol{m}^{(t=0)} = \boldsymbol{\nu}^{(t=0)} = \mathbf{0}$. To prevent the bias induced by the initial $\boldsymbol{m}^{(t=0)}$ and $\boldsymbol{\nu}^{(t=0)}$, the outputs of the two EMAs are corrected into $\hat{\boldsymbol{m}}^{(t)}$ and $\hat{\boldsymbol{\nu}}^{(t)}$. Those are used to compute the final weight update, scaled by the learning rate.

**How far to look into the past?** A typical value for $\beta_1$ is 0.9. Fig. 3a shows the weights given to past gradients for different values of $\beta$. The larger the $\beta$, the more uniform the average is. To put this in perspective—observing that $\sum_{i=0}^{\infty} \beta^i(1-\beta) = 1$ for $\beta \in [0, 1[$—the number of successive previous

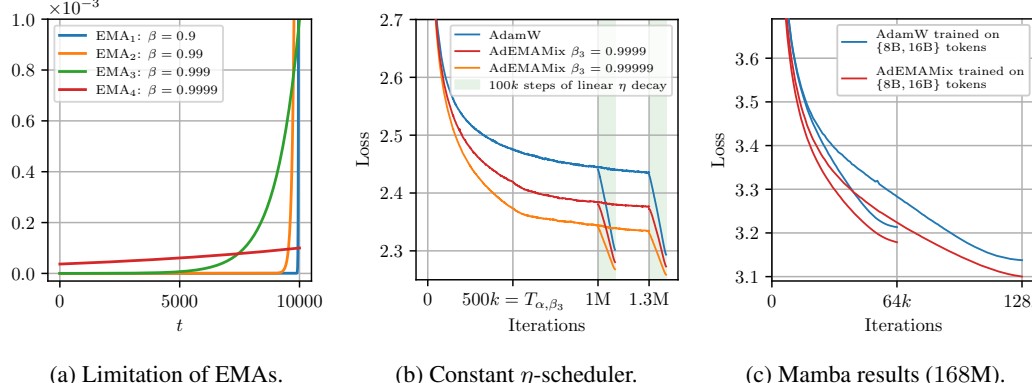

(a) Limitation of EMAs.  (b) Constant $\eta$-scheduler.  (c) Mamba results (168M).

Figure 3: **Limitation of EMAs, constant $\eta$-scheduler, & Mamba results.** In **(a)**, we plot the weights $w_t$—for each past gradient $\boldsymbol{g}^{(t)}$—given by different EMAs after $10k$ steps. For a given $\beta$, $\text{EMA}(\beta, \boldsymbol{g}^{(0)}, \dots, \boldsymbol{g}^{(T)}) = \sum_{i=0}^{T} \beta^i (1-\beta) \boldsymbol{g}^{(T-i)}$, which decays the contribution of past gradients exponentially. A small $\beta$ (e.g. 0.9) will give a high weight to the immediate past and negligible weights to older timesteps. In contrast, a high $\beta$ (e.g. 0.9999) is giving a relatively uniform, yet non-negligible weight to all gradients. No $\beta$ value can simultaneously give a high weight to the immediate past and a non-negligible weight to very old timesteps. In **(b)**, we train multiple $1.3B$ language models using $3k$ steps of warmup and then a constant learning rate $\eta = 10^{-4}$. This allows us to observe the gap between AdamW and AdEMAMix without the cosine decay as a confounder. We still use schedulers for $\alpha$ and $\beta_3$ with $T_{\alpha,\beta_3} = 500k$, $\alpha = 5$. Similar to Zhai et al. (2022); Hu et al. (2024); Hägele et al. (2024), we decay the learning rate linearly at $t = 1M$ and $t = 1.3M$. The loss-gap between AdamW and AdEMAMix increases at first, and then remains constant. AdEMAMix still outperforms AdamW after decaying the learning rate. See App.C.1.10 to see the impact of the linear decay duration. In **(c)**, we train 168M parameter Mamba models, showing how AdEMAMix's performances can generalize outside of the Transformer architecture.

steps receiving a cumulative weight of 0.5, is $t_{half} = \frac{\ln(0.5)}{\ln(\beta)} - 1$. For $\beta = 0.9$, $t_{half} \approx 6$, meaning that half of the weight is given to the previous six gradients. This observation can also be extended to SGD with e.g. polyak or nesterov momentums (Polyak, 1964; Nesterov, 1983), which typically relies on similar $\beta$ values. The value of $\beta_1$ is rarely increased beyond $\sim 0.9$. In our experiments with AdamW, increasing $\beta_1$ further degraded the performance (see App. C.1.7). Does this mean older gradients are outdated? We show that this is not the case, rather, increasing beta is reducing the sensitivity to recent gradients too much. We design AdEMAMix such that the sensitivity to recent gradients is kept, while also incorporating information from much older gradients using an additional momentum term. This allows for the use of much larger $\beta$ values e.g. 0.9999. To compare, for $\beta = 0.9999$, $t_{half} \approx 6,930$, spreading half of the mass over the previous 6,930 past gradients.

**AdEMAMix.** To keep a high sensitivity to recent gradients, while also incorporating information from older gradients, we add a second EMA (changes compared to AdamW are in Blue):

$$\begin{cases} \boldsymbol{m}_1^{(t)} = \beta_1 \boldsymbol{m}_1^{(t-1)} + (1-\beta_1)\boldsymbol{g}^{(t)}, & \hat{\boldsymbol{m}}_1^{(t)} = \frac{\boldsymbol{m}_1^{(t)}}{1-\beta_1^t} \\ \boldsymbol{m}_2^{(t)} = \beta_3 \boldsymbol{m}_2^{(t-1)} + (1-\beta_3)\boldsymbol{g}^{(t)} \\ \boldsymbol{\nu}^{(t)} = \beta_2 \boldsymbol{\nu}^{(t-1)} + (1-\beta_2)\boldsymbol{g}^{(t)2}, & \hat{\boldsymbol{\nu}}^{(t)} = \frac{\boldsymbol{\nu}^{(t)}}{1-\beta_2^t} \\ \boldsymbol{\theta}^{(t)} = \boldsymbol{\theta}^{(t-1)} - \eta\left(\frac{\hat{\boldsymbol{m}}_1^{(t)} + \alpha \boldsymbol{m}_2^{(t)}}{\sqrt{\hat{\boldsymbol{\nu}}^{(t)}} + \epsilon} + \lambda \boldsymbol{\theta}^{(t-1)}\right). \end{cases} \quad \text{(AdEMAMix)}$$

In our experiments, while the values of $\beta_1, \beta_2$ remain similar to those of equation AdamW, we often use $\beta_3 = 0.9999$. We find $\alpha \in [4, 10]$ to work well in practice.

**Tackling early training instabilities.** Early training instabilities are commonplace when training deep models, and empirically motivated the introduction of methods such as learning rate warmup and gradient clipping. Gilmer et al. (2022) show how the use of learning rate warmup can be justified from a curvature perspective; allowing the iterates to move to parts of the optimization landscape where larger learning rates are stable. While we use learning rate warmup in all our experiments,

we still noticed AdEMAMix models using a large $\beta_3$ would diverge early. This, despite not using bias correction over $\boldsymbol{m}_2$, which lets the momentum buffer fill itself slowly. Those failed runs are characterized by updates of large magnitudes in the early phase of training (see App. C.1.10, Fig. 28). For this reason, we progressively increase the values of $\beta_3$ and $\alpha$ using schedulers. For $\alpha$ we use a linear scheduler. A linear scheduler for $\beta_3$ would be ill-fitted as the same increment of $\beta_3$ has a different impact for different values of $\beta_3$. For instance, observe that an increase of $\beta$ of $\delta_\beta = 0.0001$ barely increases the $t_{half}$ for $\beta = 0.9$, while $0.999 \rightarrow 0.999 + \delta_\beta$ increases the $t_{half}$ of 77. For this reason, we design the $\beta_3$ scheduler to *increase $t_{half}$ linearly* (see App. A.1 for a derivation of that scheduler). The two schedulers are summarized below:

$$\alpha^{(t)} = f_\alpha(t, \alpha, T_\alpha) = \min(\frac{t\alpha}{T_\alpha}, \alpha), \tag{$f_\alpha$}$$

$$\beta_3^{(t)} = f_{\beta_3}(t, \beta_3, \beta_{start}, T_{\beta_3}) = \min\big( \exp\big(\frac{\ln(\beta_{start})\ln(\beta_3)}{(1 - \frac{t}{T_{\beta_3}})\ln(\beta_3) + \frac{t}{T_{\beta_3}}\ln(\beta_{start})}\big), \beta_3\big). \tag{$f_{\beta_3}$}$$

With $T_\alpha$ and $T_{\beta_3}$ are resp. the warmup times for $\alpha^{(t)}$ and $\beta_3^{(t)}$ to reach their final and maximal values. In practice we always set those two to the same value: $T_\alpha = T_{\beta_3} = T_{\alpha,\beta_3}$, and we typically use $T_{\alpha,\beta_3} = T$, with $T$ being the total number of iterations. $\beta_{start}$ is always set to $\beta_1$ in our experiments. The use of those schedulers is not always required, especially, we found those have no impact when AdEMAMix is activated later during training (see Fig. 5). The full AdEMAMix optimizer, including the schedulers, is shown in Alg. 1.

---

**Algorithm 1** AdEMAMix optimizer. Differences with AdamW are in blue.

---

1: **Input:** Data distribution $\mathcal{D}$. Initial model parameters $\boldsymbol{\theta}^{(0)}$. Number of iterations $T$. Learning rate $\eta$. $\epsilon$ a small constant. AdamW parameters: $\beta_1$, $\beta_2$ and $\lambda$. AdEMAMix parameters $\beta_3$, $\alpha$. Warmup parameter $T_{\alpha,\beta_3}$, note that we usually set it to $T$. $\beta_{start}$ is usually set to $\beta_1$.
2: Initialize timestep: $t \leftarrow 0$
3: Initialize EMAs: $\boldsymbol{m}_1^{(0)} \leftarrow \boldsymbol{0}$ , $\boldsymbol{m}_2^{(0)} \leftarrow \boldsymbol{0}$ , $\boldsymbol{\nu}^{(0)} \leftarrow \boldsymbol{0}$
4: **for** $t \in [T]$ **do**
5:  $t \leftarrow t + 1$
6:  Optional: use schedulers $\eta^{(t)}, \beta_3^{(t)} \leftarrow f_{\beta_3}(t, \beta_3, \beta_{start}, T_{\alpha,\beta_3})$ and $\alpha^{(t)} \leftarrow f_\alpha(t, \alpha, T_{\alpha,\beta_3})$
7:  Sample batch: $x \sim \mathcal{D}$
8:  Compute gradient: $\boldsymbol{g}^{(t)} \leftarrow \nabla_{\boldsymbol{\theta}}\mathcal{L}_{\boldsymbol{\theta}^{(t-1)}}(\boldsymbol{x})$
9:  Update the fast EMA $\boldsymbol{m}_1$: $\boldsymbol{m}_1^{(t)} \leftarrow \beta_1\boldsymbol{m}_1^{(t-1)} + (1 - \beta_1)\boldsymbol{g}^{(t)}$
10:  Update the slow EMA $\boldsymbol{m}_2$: $\boldsymbol{m}_2^{(t)} \leftarrow \beta_3^{(t)}\boldsymbol{m}_2^{(t-1)} + (1 - \beta_3^{(t)})\boldsymbol{g}^{(t)}$
11:  Update the second moment estimate: $\boldsymbol{\nu}^{(t)} \leftarrow \beta_2\boldsymbol{\nu}^{(t-1)} + (1 - \beta_2)\boldsymbol{g}^{(t)2}$
12:  Apply bias corrections: $\hat{\boldsymbol{m}}_1^{(t)} \leftarrow \frac{\boldsymbol{m}_1^{(t)}}{1-\beta_1^t}$ , $\hat{\boldsymbol{\nu}}_1^{(t)} \leftarrow \frac{\boldsymbol{\nu}_1^{(t)}}{1-\beta_2^t}$
13:  Update parameters: $\boldsymbol{\theta}^{(t)} \leftarrow \boldsymbol{\theta}^{(t-1)} - \eta^{(t)}\big(\frac{\hat{\boldsymbol{m}}_1^{(t)} + \alpha^{(t)}\boldsymbol{m}_2^{(t)}}{\sqrt{\hat{\boldsymbol{\nu}}^{(t)}}+\epsilon} + \lambda\boldsymbol{\theta}^{(t-1)}\big)$
14: **end for**
15: **Return** optimized parameters $\boldsymbol{\theta}^{(T)}$

---

**Using $\beta_1 = 0$ to save memory.** By setting $\beta_1 = 0$, we can save on memory by replacing $\hat{\boldsymbol{m}}_1^{(t)}$ by $\boldsymbol{g}^{(t)}$. In this case, the memory cost of AdEMAMix is the same as for AdamW. We show in App. C.1.5 (Fig. 16b) and App. C.1.8 that using $\beta_1 = 0$ often works, at the cost of potentially less stable training.

**Computational overheads.** Adding an additional EMA requires additional memory and compute. The added compute is negligible in comparison to what is required for the forward-backward, and has little impact over the total runtime as shown in Fig. 5a. Moreover—when considering larger distributed setups—it is worth noting that AdEMAMix is not increasing communication (gradient reduction) over Adam. Therefore, we expect the runtime overhead of AdEMAMix to shrink in those settings, as data movements occupy a larger fraction of the total runtime. A more significant overhead is in terms of memory when $\beta_1 \neq 0$, as AdEMAMix requires to allocate both $\boldsymbol{m}_1$ and $\boldsymbol{m}_2$, which are of the same size as the model parameters $\boldsymbol{\theta}$. We believe this issue is of lesser consequences as Fully-Sharded-Data-Paralellism (Zhao et al., 2023, FSDP) can always be used to distribute the optimizer states across compute nodes. As an example, using the Megatron library (Shoeybi et al., 2020) to

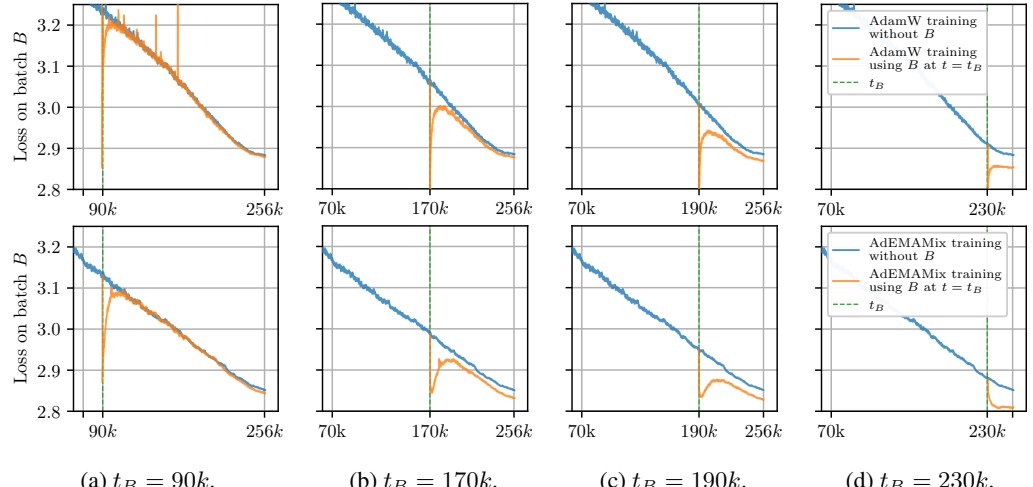

Figure 4: **Measuring forgetting using a held-out batch** $B$**.** The top row is for AdamW, the bottom row is for AdEMAMix. We trained one AdamW and AdEMAMix model on a RedPajama dataset *not* containing the batch $B$, those runs are in blue. We then run multiple experiments where we inject $B$ in the training data at a specific timestep $t_B$. Those runs are in orange. To inspect how much influence $B$ had when it is injected at $t_B$, we can observe the evolution of the gap between the blue and the orange curves. For both optimizers, we observe a rapid decrease of the loss on $B$ right after training on $B$. The sharpness of this decrease in loss is more pronounced for AdamW compared to AdEMAMix. However, when using AdamW, the loss on $B$ then increases faster, which we interpret as the model forgetting $B$ faster. In contrast, the curves for AdEMAMix are smoother, the loss on $B$ goes back up slower, and ultimately $B$ had a bigger impact on the training when using AdEMAMix—as can be seen by looking at the larger gap between the orange and blue curves for the last iteration. Finally, the forgetting behaviour evolves during training, with later training batches being remembered better. See App. C.1.3 for more experiments on forgetting.

train an 8B model across 16 compute nodes, each having 8 GPUs. Assuming a data-parallelism (DP) of 64 (2 GPUs per replica). This would imply sharding the optimizer states over the 64 replicas, causing a small *per-GPU* offset of 0.5GB when compared with AdamW. In the cases where memory remains an issue, one mitigation strategy could be to apply factorization strategies as in Shazeer & Stern (2018); Zhao et al. (2024) , or reduce the size of the momentum buffers as in Zhang et al. (2025).

**Hyperparameter sensitivity.** While we introduce up to four new hyperparameters: $\alpha$, $\beta_3$, $T_\alpha$, and $T_{\beta_3}$, in practice we always set $T_\alpha = T_{\beta_3} = T_{\alpha,\beta_3}$, and use $T_{\alpha,\beta_3} = T$ in most cases. We show in App. C.1.5 that $T_\alpha$ and $T_{\beta_3}$ should only be large enough to prevent instabilities early during training. While all of our experiments on language modeling use $\beta_3 = 0.9999$, other values such as $0.999$ or even $0.99999$ still can outperform the AdamW baseline (see App. C.1.5, Fig. 13). On vision tasks, the scatter plots in Fig. 6 show all the AdEMAMix models trained for those experiments, highlighting how easy it can be to find good $\alpha$ and $\beta_3$ values. Overall, we find the ranges of values of $\alpha$, $\beta_3$ and $T_{\alpha,\beta_3}$ providing improvements over AdamW to be wide. See App. C.1.5 for hyperparameter sweeps.

**Gradient clipping &** $\beta_2$**.** Adam can diverge in cases where $\beta_2$ is not sufficiently large compared to $\beta_1$ (Zhang et al., 2022). Intuitively, the denominator in the AdamW update can shrink faster than the numerator, causing a spike in update magnitude . Given the proximity of our method to AdamW, a similar observation applies to AdEMAMix, and we require $\beta_2$ to be large enough. In practice, we observe that both $\beta_2 = 0.999$ or $0.9999$ work well. Moreover, we notice that gradient clipping mitigates the issue and we use small gradient clipping values in our experiments (e.g. $0.5$ or $0.1$). We also believe clipping to help dealing with outlier gradients, which can have a long-lasting and detrimental effect for AdEMAMix given the large $\beta_3$.

**Limitations.** AdEMAMix consists in leveraging very old gradients. Therefore, the method is best suited to settings where the number of iterations is important. We report on this effect in App. C.1.6,

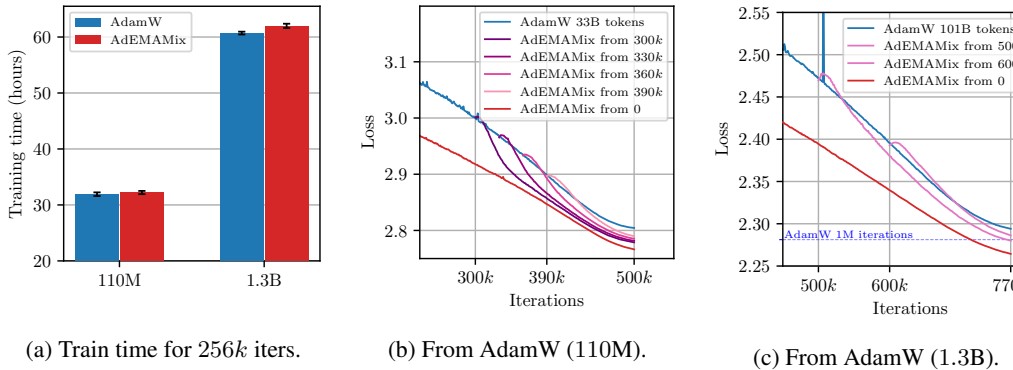

(a) Train time for $256k$ iters.  (b) From AdamW (110M).  (c) From AdamW (1.3B).

Figure 5: **Training time comparison & starting AdEMAMix from AdamW.** In **(a)**, we compare the time required to train 110M and 1.3B parameter models for $256k$ iterations. The additional EMA renders AdEMAMix slightly slower than AdamW. However, if we were to train AdamW longer to compensate for this gap, we would only train for an additional $2379$ and $5421$ iterations for resp. 110M and 1.3B parameter models. Those additional iterations would not be sufficient to close the gap (see Fig. 1). In **(b)** and **(c)**, we show—for two different model sizes—the effect of switching from AdamW to AdEMAMix during training. AdEMAMix's additional parameter $m_2$ is initialized to $0$, no scheduler is used for $\alpha$ or $\beta_3$. For both model sizes, we observe the loss increases slightly at first before decreasing and outperforming the baseline. In both cases, the earlier AdEMAMix is used, the better the final loss. See App. C.1.10 for results using schedulers.

additionally showing how smaller values of $\beta_3$ (e.g. $\beta_3 = 0.999$) can be better for low iterations scenarios. Moreover, retaining gradient information over many thousands steps can pose a problem in domains requiring fast adaptation to a sudden distribution shift.

## 4 RESULTS

In this section we use AdEMAMix to train language models (§ 4.1 and § 4.2) and vision transformers (§ 4.3) ranging from 24M to 1.3B parameters.

### 4.1 TRANSFORMER LLM TRAINING

**Experimental setup.** We use a transformer architecture (Vaswani et al., 2017) with learnt positional encoding. All of our experiments are using sequences of 1,024 tokens. We experiment with three sizes of transformers: 110M, 335M, and 1.3B. For the learning rate, we use 3k warmup steps followed by—unless specified—a cosine decay to $10^{-5}$. We extensively tuned the hyperparameters for both AdamW and AdEMAMix models (see App. B.1). We use the RedPajama v2 (Computer, 2023) dataset for all of our experiments. We use batch sizes of 64, 96 and 128 for respectively our 110M, 335M, and 1.3B parameter models. Depending on the model, we vary the number of iterations from $256k$ to 1.5M. For AdEMAMix, we use $\beta_3 = 0.9999$ and $\alpha \in \{5, 8, 10\}$ depending on the model. A full description of the architecture and hyperparameters used is in App. B.1. We train on up to 8 A100 NVIDIA GPUs using data-parallelism.

**Why not simply increasing AdamW's $\beta_1$?** While our toy experiment in Fig. 2 already gives some intuition on why increasing Adam's $\beta_1$ is likely not to have the same effect as having an additional EMA as in AdEMAMix, we verify this intuition by training multiple 110M models using Adam with large $\beta_1 \in \{0.99, 0.999, 0.9999, 0.99999\}$. When we use a large $\beta_1$ from the beginning of training, we observe instabilities for larger $\beta_1$ values and no $\beta_1 > 0.9$ improves upon the AdamW baseline. One can imagine this to be due to increasing $\beta_1$ too early. Therefore, we also modify AdamW and add the same scheduler on $\beta_1$ as we use on AdEMAMix's $\beta_3$. $\beta_1$ is now increased steadily over the entire training duration. While this mostly stabilizes the training, none of the experiments outperformed the baseline using $\beta_1 = 0.9$. Moreover, to rule out any effect that could be due to early training instabilities, we run the same experiments starting from a pre-trained AdamW checkpoint trained for $300k$ iterations. We simply resume training and either increase $\beta_1$ suddenly or using a scheduler.

Here again—unlike when using AdEMAMix—none of those experiments outperform the baseline. The details of those experiments are available in App. C.1.7. Those experiments show that simply increasing the $\beta_1$ value in AdamW is not enough, which justifies our design of AdEMAMix.

**Better perplexity for the same number of steps.** For all model sizes, all the number of iterations used—ranging from 256k to 1M—, AdEMAMix always outperforms the AdamW baseline. In Fig. 1, we show the validation loss curves for AdamW and AdEMAMix models trained—for each size—on various numbers of tokens. For 110M parameter models, training for $256k$ iterations gives similar results as training an AdamW model for $500k$ iterations. The gap between baseline and our method seems to be increasing as we increase the number of iterations. For 1.3B parameter models, training using $770k$ steps is on par with training the baseline for 1.5M iterations—reaching the same performance with 51% fewer tokens (an economy of 96M tokens). In Fig. 3b, we observe similar improvements when using a constant and then linear decay learning rate scheduler.

**Training speed comparison.** We measure the time per iteration for all of our experiments. In Fig. 5a we plot the time required to train our 110M and 1.3B parameter models for $256k$ iterations. We observe that the impact of using an additional EMA on the training speed is negligible. If we were to train new models with a fix time budget, the extra iterations of the baseline would not be sufficient to close the gap. For instance, to match a 110M parameter AdEMAMix model trained for $256k$ iterations, we need to train an AdamW model for $500k$ iterations, and the time advantage of AdamW would only allow us to do 2,379 additional iterations. Moreover, as mentioned in § 3, we expect the time overhead to decrease when multi-node training is used, as IOs would become an important bottleneck, and AdEMAMix is not increasing IOs.

**Consistency of the gain across token-budgets.** In Fig. 1, given that enough tokens have been used w.r.t. the model size, we observe consistent gains accross token budgets. It seems our method is able to bring a constant improvement over the baseline. This can be seen more clearly in Fig. 3b, showing results when using a constant learning rate scheduler—which removes the confounder of the cosine learning rate decay. We observe how, after an initial phase in which the gap grows, this gap becomes seemingly constant after a sufficient number of iterations.

**Resume from AdamW vs. training AdEMAMix from scratch.** So far we trained AdEMAMix models from scratch. We show it is also possible to switch from an AdamW to an AdEMAMix state in the middle of training. When switching to AdEMAMix at step $t_{switch}$, we initialize $\boldsymbol{m}_2^{(t_{switch})} = \boldsymbol{0}$ and replace $t$ by $t - t_{switch}$ in the scheduler equations—if those are used. However, we find that schedulers are not required when resuming training and report results without them in the main paper (see App. C.1.10 for more details). In Fig 5b and Fig.5c, we show that (i) it is possible to improve upon the baseline when activating AdEMAMix later during training, and (ii) the earlier the switch, the larger the gain, with diminishing returns. This indicates that the improvement of AdEMAMix cannot be attributed solely to early training dynamics, but rather, late training dynamics seem to play an important role. This is further corroborated by the reverse experiment—which switches from AdEMAMix to AdamW mid-training and show a performance degradation (see results in App. C.1.4).

**AdEMAMix models forget the training data slower.** As an attempt to understand the reason for AdEMAMix's improvements over AdamW, we study how fast a training batch is forgotten after being used during training. We focus on following one specific batch $B$. We start by removing $B$ from the RedPajama training data and train AdamW and AdEMAMix models. For those runs, $B$ is therefore akin to a batch from the validation set. We measure the loss on $B$ while training. Now we can resume training from various checkpoints, and inject $B$ into the training data at various times $t_B$. By comparing the two runs—one having trained on $B$, while the other never saw $B$—we can visualize how $B$ is learned, and how it is forgotten. After training on $B$, we expect the loss on $B$ to decrease suddenly and then increase as the model forgets the contribution of that batch. When comparing the forgetting curves for AdamW and AdEMAMix in Fig. 4, we see striking differences. AdamW models forget much faster—the loss over $B$ increases faster—than AdEMAMix models. Moreover, at the end of training, batches processed by AdEMAMix see their loss being improved over many thousands of iterations. Additional experiments on forgetting can be found in App. C.1.3.

## 4.2 MAMBA LM TRAINING

**Experimental setup & results.** Our setup is similar to § 4.1, except that we now train 168M parameter Mamba models (Gu & Dao, 2023) using the FineWeb dataset (Penedo et al., 2024). See

App. B.2 for details. In Fig. 3c the improvement using AdEMAMix is consistent with our experiments on Transformer models, showing how AdEMAMix's gains can extend beyond Transformers.

## 4.3 ViT Training

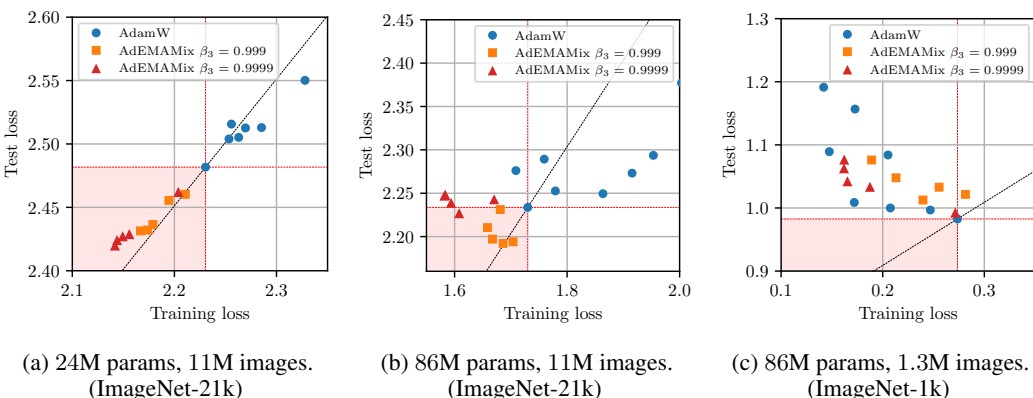

(a) 24M params, 11M images.
(ImageNet-21k)

(b) 86M params, 11M images.
(ImageNet-21k)

(c) 86M params, 1.3M images.
(ImageNet-1k)

Figure 6: **ViT results for different capacity/data ratio.** As described in § 4.3, we compare the train vs. test loss between AdEMAMix and AdamW, on multiple scenarios **(a,b,c)**. Those have different capacity/data ratios. For each setting, we first tune an AdamW baseline by testing multiple hyperparameters, those are represented by blue dots. We then pick the hyperparameters of the best baseline—corresponding to the lowest test loss—and train multiple AdEMAMix models by simply testing various $\alpha \in \{1, 5, 10, 15, 20\}$ and $\beta_3 \in \{0.999, 0.9999\}$. Those are represented by orange squares and red triangles. See App. B.3 for detailed hyperparameters. The red area in the above plots represents the area in which we achieve both a better training and test loss compared to the best baseline. The diagonal black line represents the line we would follow when we improve upon the baseline without adding any overfitting (i.e. an improvement in train loss yields the same improvement in test loss). **(a)** represents the most desirable setting, in which we have relatively a lot of data compared to the model size. In this setting, AdEMAMix outperforms the baseline for all hyperparameters tested. If we increase the model size as in **(b)**, most AdEMAMix hyperparameters are improving upon the baseline. If we now decrease the dataset size, using a 86M model with 1.3M images as in **(c)**, doing a total of 320 epochs, we do not observe an improvement from using AdEMAMix. See App. C.2 to see classification accuracies and loss curves for those experiments.

**Experimental setup.** In this section we consider a different setting in which the data is now a limited resource, and we do multiple epochs (e.g. 37 or 320). We use two subsets of the ImageNet (Russakovsky et al., 2015) dataset: (i) the widely used ImageNet-1k subset, consisting of 1.3M images and 1,000 possible classes; (ii) a filtered and pre-processed version of the ImageNet-21k (Ridnik et al., 2021) containing 11M images corresponding to 10,450 classes. For each, we measure the test loss on a held-out test set. We use the ViT architecture (Dosovitskiy et al., 2021b) at two different scales: 24M and 86M parameters. Importantly, if it is common in the vision literature to pre-train large ViT models and finetune them on smaller datasets, this work focuses on pretraining optimization and we therefore train and test on the same distribution. The models' hyperparameters are detailed in App. B.3, we use a batch size of 4096 for all our experiments. We used training hyperparameters from Dosovitskiy et al. (2021b) as a starting point and did some additional tuning of the learning-rate, dropout, and weight decay for our AdamW baselines. We then use the hyperparameters of the best AdamW baseline and train AdEMAMix models with various values of $\alpha \in \{1, 5, 10, 15, 20\}$ and $\beta_3 \in \{0.999, 0.9999\}$. We train models for 320 and 37 epochs for resp. the ImageNet-1k and ImageNet-21k datasets, corresponding in both cases to $100k$ iterations. Data-augmentation techniques have been shown to be central to the efficient training of ViTs Touvron et al. (2021; 2022). We use simple data-augmentations, which includes mixup (Zhang et al., 2018). We train on 8 A100 NVIDIA GPUs using Pytorch Fully Sharded Data-Parallelism (Zhao et al., 2023, FSDP).

**AdEMAMix for different capacity/data ratios.** We consider three scenarios that differ by their capacity/data ratios. First, we trained 24M parameter models on 11M images (ImageNet-21k), for 37 epochs. In this setting, as can be seen in Fig. 6a, it is trivial to find AdEMAMix parameters

outperforming the baseline in terms of both training and test accuracy. We now increase the model size to 86M parameters. Fig. 6b shows it is still easy to find parameters outperforming the baseline. We now keep the model size of 86M parameters and switch to the smaller ImageNet-1k dataset (1.3M images), which—given our $100k$ iterations—increases the number of epochs to 320. In this setting, Fig. 6c shows that outperforming the baseline becomes difficult. AdEMAMix seems to perform best in scenarios with large volumes of data w.r.t. the capacity of the model.

## 5 CONCLUSION

In this work, we find that old gradients can be leveraged to efficiently train large language models and ViTs. Our proposed optimizer combines two momentum terms. A slow (large $\beta$) momentum gathers information over many timestep, while a fast (slow $\beta$) momentum can adapt the trajectory of the iterates to the rapidly changing loss landscape. We demonstrate the superiority of our optimizer over AdamW through a set of experiments on text modeling and image classification. We moreover reveal how our optimizer forgets the training data at a slower pace.

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

# Contents

# A  IMPLEMENTATION DETAILS

## A.1  DERIVING THE $\beta_3$ SCHEDULER

Let's consider $S(t)$, the sum of the weights given to the last $t$ gradients by an EMA parameterized by $\beta \in [0, 1[$:

$$S(t) = (1 - \beta) \sum_{i=0}^{t} \beta^i$$

We want to know which timestep $t$ would correspond to a cumulative weight of $0.5$:

$$(1 - \beta) \sum_{i=0}^{t} \beta^i = 0.5 \Leftrightarrow \beta^{t+1} = 0.5 \Leftrightarrow t = \frac{\ln(0.5)}{\ln(\beta)} - 1$$

Let $f(\beta) = \frac{\ln(0.5)}{\ln(\beta)} - 1$. This function provides how many past consecutive gradients receive a cumulative weight of $0.5$.

Its inverse is:

$$f^{-1}(t) = 0.5^{\frac{1}{t+1}}$$

We want a scheduler which increases $\beta$ from $\beta_{start}$ to $\beta_{end}$ such that $f(\beta)$ increases linearly. Given an interpolating parameter $\mu \in [0, 1]$, this scheduler can be written as:

$$\beta(\mu) = f^{-1}((1 - \mu)f(\beta_{start}) + \mu f(\beta_{end}))$$

By replacing $f$ and $f^{-1}$ by their respective formula, one can arrive to:

$$\beta(\mu) = \exp \left( \frac{\ln(\beta_{start}) \ln(\beta_{end})}{(1 - \mu) \ln(\beta_{end}) + \mu \ln(\beta_{start})} \right)$$

By setting $\beta_{end} = \beta_3$ and $\mu = \frac{t}{T_{\beta_3}}$, we arrive to the $\beta_3$-scheduler introduced in § 3. We show the shape of our scheduler and compare it to a linear scheduler in Fig. 7.

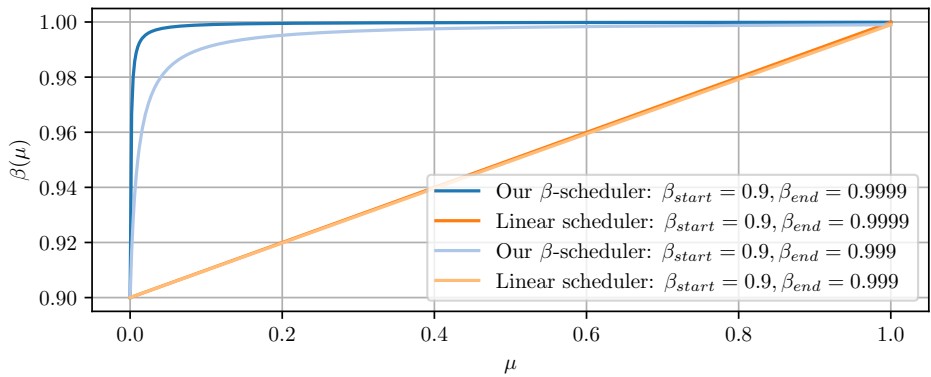

Figure 7: **AdEMAMix's $\beta_3$ scheduler.** We compare our scheduler to a linear scheduler for $\beta_{start} = 0.9$ and $\beta_{end} \in \{0.999, 0.9999\}$. While our scheduler looks more aggressive at first glance, it increases fast for smaller values of $\beta$, and slowly for larger ones associated. This makes sense as the same increase of $\beta$ for larger $\beta$ values has a bigger impact than that same increase applied to a smaller value of $\beta$. The two linear schedulers look practically the same, despite values of $\beta_{end}$ differing by one order of magnitude. This is not the case with our scheduler.

## A.2 Pytorch implementation

The following is a code skeleton for our AdEMAMix optimizer in Pytorch (Paszke et al., 2019). The full implementation of AdEMAMix in Pytorch is provided in the following repository: https://github.com/apple/ml-ademamix

Listing 1: AdEMAMix code skeleton using Pytorch

```python
import math
import torch
from torch.optim import Optimizer

class AdEMAMix(Optimizer):

    def __init__(self,
                 params,
                 lr=1e-3,
                 betas=(0.9,0.999,0.9999),
                 alpha=5.0,
                 T_beta3=0,
                 T_alpha=0,
                 eps=1e-8,
                 weight_decay=0.0):
        # init the optimizer

    @torch.no_grad()
    def step(self):

        for group in self.param_groups:

            lr = group["lr"]
            lmbda = group["weight_decay"]
            eps = group["eps"]
            beta1, beta2, beta3_final = group["betas"]
            T_beta3 = group["T_beta3"]
            T_alpha = group["T_alpha"]
            alpha_final = group["alpha"]

            for p in group["params"]:

                grad = p.grad
                state = self.state[p]

                # State initialization
                if len(state) == 0:
                    state["step"] = 0 # step counter used for bias correction
                    state["m1"] = torch.zeros_like(p) # fast EMA
                    state["m2"] = torch.zeros_like(p) # slow EMA
                    state["nu"] = torch.zeros_like(p) # second moment estimate

                m1, m2, nu = state["m1"], state["m2"], state["nu"]

                # Bias correction: no correction for beta3's EMA
                state["step"] += 1
                bias_correction1 = 1 - beta1 ** state["step"]
                bias_correction2 = 1 - beta2 ** state["step"]

                # Calling the schedulers for alpha and beta3
                alpha = alpha_scheduler(state["step"], start=0, end=alpha_final, T=T_alpha)
                beta3 = beta3_scheduler(state["step"], start=beta1, end=beta3_final, T=T_beta3)

                # Update the EMAs
                m1.mul_(beta1).add_(grad, alpha=1 - beta1)
                m2.mul_(beta3).add_(grad, alpha=1 - beta3)
                nu.mul_(beta2).addcmul_(grad, grad, value=1 - beta2)

                # Compute step
                denom = (nu.sqrt() / math.sqrt(bias_correction2)).add_(eps)
                update = (m1.div(bias_correction1) + alpha * m2) / denom

                # Add weight decay
                update.add_(p, alpha=lmbda)

                # Apply the update scaled by the learning rate
                p.add_(-lr * update)

        return loss
```

## A.3 Optax implementation

The following is a code skeleton for our AdEMAMix optimizer in Optax, an optimization library based on Jax (Bradbury et al., 2018). The full implementation of AdEMAMix in Jax is provided in the following repository: https://github.com/apple/ml-ademamix

Listing 2: AdEMAMix code skeleton using JAX and Optax

```python
from typing import NamedTuple
import chex
from optax._src import transform, combine, base, numerics
from jax import tree_util as jtu
import jax.numpy as jnp

class ScaleByAdemamixState(NamedTuple):
  count: chex.Array
  count_m2: chex.Array
  m1: base.Updates
  m2: base.Updates
  nu: base.Updates

def ademamix(lr,
             b1=0.9,
             b2=0.999,
             b3=0.9999,
             alpha=5.0,
             b3_scheduler=None,
             alpha_scheduler=None,
             eps=1e-8,
             weight_decay=0.0):
  return combine.chain(
    scale_by_ademamix(b1, b2, b3, alpha, b3_scheduler, alpha_scheduler, eps),
    transform.add_decayed_weights(weight_decay),
    transform.scale_by_learning_rate(lr),
  )

def scale_by_ademamix(b1,
                      b2,
                      b3,
                      alpha,
                      b3_scheduler,
                      alpha_scheduler,
                      eps):

  def init_fn(params):
    m1 = tree_zeros_like(params)  # fast EMA
    m2 = tree_zeros_like(params)  # slow EMA
    nu = tree_zeros_like(params)  # second moment estimate
    return ScaleByAdemamixState(
        count=jnp.zeros([], jnp.int32),
        count_mu2=jnp.zeros([], jnp.int32),
        m1=m1,
        m2=m2,
        nu=nu
    )

  def update_fn(updates, state, params=None):
    del params
    c_b3 = b3_scheduler(state.count_m2) if b3_scheduler is not None else b3
    c_alpha = alpha_scheduler(state.count_m2) if alpha_scheduler is not None else alpha
    m1 = tree_update_moment(updates, state.m1, b1, 1) # m1 = b1 * m1 + (1-b1) * updates
    m2 = tree_update_moment(updates, state.m2, c_b3, 1)
    nu = tree_update_moment_per_elem_norm(updates, state.nu, b2, 2)
    count_inc = numerics.safe_int32_increment(state.count)
    count_m2_inc = numerics.safe_int32_increment(state.count_m2)
    m1_hat = tree_bias_correction(m1, b1, count_inc)
    nu_hat = tree_bias_correction(nu, b2, count_inc)
    updates = jtu.tree_map(
        lambda m1_, m2_, v_: (m1_+c_alpha*m2_)/(jnp.sqrt(v_)+eps),
        m1_hat,
        m2,
        nu_hat
    )
    return updates, ScaleByAdemamixState(
        count=count_inc,
        count_m2=count_m2_inc,
        m1=m1,
        m2=m2,
        nu=nu
    )

  return base.GradientTransformation(init_fn, update_fn)
```

## B EXPERIMENTAL DETAILS

### B.1 TRANSFORMER LLM EXPERIMENTS

**Architecture details.** Our architecture is based on the transformer decoder of Vaswani et al. (2017). We use learnt positional encoding. We use a SentencePiece (Kudo & Richardson, 2018) tokenizer with a vocabulary of 32000 tokens. The model specific parameters for the different sizes are summarized in Table 1. Our implementation is using Jax (Bradbury et al., 2018), and we train using `bfloat16`, except for normalization modules and softmax which use `float32`. The optimizer states and operations are in `float32`.

Table 1: **Model parameters for our LLM experiments.**

| Model parameters | 110M | 330M | 1.3B |
|---|---|---|---|
| Hidden size | 768 | 1024 | 2048 |
| FFW expansion factor | 4 | 4 | 4 |
| Attention heads | 12 | 16 | 16 |
| Layers | 12 | 24 | 24 |

**How did we tune the hyperparameters?** Starting from our smallest models (110M parameters), we first tuned hyperparameters for our AdamW models. We then use the best hyperparameters found as a starting point for AdEMAMix and tuned $\beta_3$ and $\alpha$. When we use schedulers for $\alpha$ and $\beta_3$, we set $T_\alpha = T_{\beta_3} = T$, with $T$ being the total number of iterations. Table 2 summarizes the hyperparameters we tried for this model size. The impact of AdEMAMix's hyperparameters is discussed extensively in App. C.1.5. For our 330M parameter models, we mostly kept the best hyperparameters found for the 110M model and re-tuned the learning rate and gradient clipping. For AdEMAMix, we additionally tested multiple $\beta_3$ and $\alpha$ values. We summarize this process in Table 3. Finally, for our 1.3B parameter models, we re-iterated the same process, re-tuning only the learning rate and gradient clipping parameters for our AdamW runs. When trying to transfer the best learning rate found to AdEMAMix, we found it to be too high, causing instabilities we couldn't fix using gradient clipping. For this reason, we also tuned the learning rate for AdEMAMix for this model size. This process is described in Table 4.

Table 2: **Hyperparameter tuning for our** 110**M parameter LLM models.** In this table we report the hyperparameters we tuned for our 110M parameter models. When multiple values are given, we bold the parameters we found to give the best results.

| Hyperparameter | Value |
|---|---|
| Learning rate $\eta$ | 0.005, 0.002, **0.001**, 0.0005, 0.0001 |
| Number of warmup steps | 2000, **3000**, 4000, 5000, 6000 |
| Sequence length | 1024 |
| Weight decay $\lambda$ | **0.1**, 0.0 |
| Learning rate decay scheduler | no-decay, **cosine-decay** |
| Batch size | 64 |
| Gradient clipping | None, 5.0, 1.0, **0.5** |
| AdamW $\beta_1$ | **0.9**, 0.99, 0.999, 0.9999 |
| AdamW $\beta_2$ | 0.95, **0.999** |
| AdEMAMix $\beta_1$ | 0.9 |
| AdEMAMix $\beta_2$ | 0.999 |
| AdEMAMix $\beta_3$ | 0.999, **0.9999**, 0.99999 |
| AdEMAMix $\alpha$ | 2, 4, 6, **8**, **10**, 15, 20 |

**Hyperparameters for experiments switching from AdamW and AdEMAMix.** For experiments in Fig. 5b and Fig. 5c, when we switch from AdamW to AdEMAMix during training, for our 110M parameter models (Fig. 5b), we use $\alpha = 2, \beta_3 = 0.9999$ and $T_{\alpha,\beta_3} = 0$. For our 1.3B parameter

Table 3: **Hyperparameter tuning for our** 330**M parameter LLM models.** In this table we report the hyperparameters we tuned for our 330M parameter models. When multiple values are given, we bold the parameters we found to give the best results.

| Hyperparameter | Value |
|---|---|
| Learning rate $\eta$ | 0.001, **0.0005**, 0.0001 |
| Number of warmup steps | 3000 |
| Sequence length | 1024 |
| Weight decay $\lambda$ | 0.1 |
| Learning rate decay scheduler | cosine-decay |
| Batch size | 96 |
| Gradient clipping | 1.0, 0.5, **0.1** |
| AdamW $\beta_1$ | 0.9 |
| AdamW $\beta_2$ | 0.999 |
| AdEMAMix $\beta_1$ | 0.9 |
| AdEMAMix $\beta_2$ | 0.999 |
| AdEMAMix $\beta_3$ | 0.999, **0.9999**, 0.99999 |
| AdEMAMix $\alpha$ | 5, **8**, 10, 15 |

Table 4: **Hyperparameter tuning for our** 1.3**B parameter LLM models.** In this table we report the hyperparameters we tuned for our 1.3B parameter models. When multiple values are given, we bold the parameters we found to give the best results.

| Hyperparameter | Value |
|---|---|
| Number of warmup steps | 3000 |
| Sequence length | 1024 |
| Weight decay $\lambda$ | 0.1 |
| Learning rate decay scheduler | cosine-decay |
| Batch size | 128 |
| Gradient clipping | 1.0, 0.5, **0.1** |
| Learning rate $\eta$ for AdamW | 0.001, **0.0005**, 0.0003, 0.0001, 0.00005 |
| AdamW $\beta_1$ | 0.9 |
| AdamW $\beta_2$ | 0.999 |
| Learning rate $\eta$ for AdEMAMix | 0.0005, **0.0003** |
| AdEMAMix $\beta_1$ | 0.9 |
| AdEMAMix $\beta_2$ | 0.999 |
| AdEMAMix $\beta_3$ | 0.999, **0.9999**, 0.99999 |
| AdEMAMix $\alpha$ | 1, 3, **5**, 8, 10, 15 |

models (Fig. 5c), we use $\alpha = 1$, $\beta_3 = 0.9999$ and $T_{\alpha,\beta_3} = 0$. Other hyperparameters are inherited from the AdamW model we are switching from.

**Hyperparameters used our constant learning rate scheduler experiments.** For Fig. 3b we use $\eta = 10^{-4}$, the remaining of the hyperparameters are identical as those used for our other 1.3B experiments and provided in Table 4.

**Hyperparameters used in our forgetting experiments.** For the experiments in Fig. 4, we used 110M parameter models and hyperparameters from Table 2.

## B.2 MAMBA LM EXPERIMENTS

**Architecture details.** We use a Mamba architecture (Gu & Dao, 2023) with an embedding dimension of 768, an expansion factor of 2, a state size of 16, and a depth of 24. We use a batch size of 120 sequences of 1024 tokens. We use the `EleutherAI/gpt-neox-20b` tokenizer from Black et al. (2022), using the HuggingFace library (Wolf et al., 2019).

**Hyperparameter details.** We used parameters mostly taken from Gu & Dao (2023):

- Learning rate: $\eta = 0.0006$,
- Warmup steps: $3k$,
- $\eta$-scheduler type: cosine decay to $10^{-5}$,
- Weight-decay: $\lambda = 0.1$,
- Gradient clipping value: 1,
- Total training steps $T \in \{64k, 128k\}$.

For experiments with AdamW, we use $\beta_1 = 0.9$ and $\beta_2 = 0.999$. For AdEMAMix, we use $\beta_1 = 0.9, \beta_2 = 0.999, \beta_3 = 0.9999, T_{\alpha,\beta_3} = T$ and $\alpha = 8$.

## B.3 ViT EXPERIMENTS

**Architecture details.** We use a ViT architecture following Dosovitskiy et al. (2021b). The architecture details for both our 24M and 86M parameter models can be seen in Table 5. We do not use an EMA of the iterates as our final model. Our implementation is in Pytorch (Paszke et al., 2019), and use `bfloat16`.

Table 5: **Model parameters for our ViT experiments.**

| Model parameters | 24M | 86M |
|---|---|---|
| Patch size | 16 | 16 |
| Number of patches | $14 \times 14$ | $14 \times 14$ |
| Image size | 224 | 224 |
| Embedding dim. | 384 | 768 |
| MLP dim. | 1536 | 3072 |
| Layers | 12 | 12 |
| Number of heads | 6 | 12 |

**Data augmentation and Mixup.** While more sophisticated augmentation methods exist (Touvron et al., 2022), we simply use random-resized cropping, random horizontal flip ($p = 0.5$), and random color jitter (applied with a probability $p = 0.8$).

**How did we tune the hyperparameters?** For each model size, we started by tuning the hyperparameters for the AdamW baseline. The hyperparameters we tuned and the values we retained for our three different settings are in Table 6. For our experiments on ImageNet-1k, given that all models overfit the dataset, we select the best model according to the minimum validation loss, akin to using early stopping. For AdEMAMix, for each setting, we use the hyperparameters of the best AdamW model and only tune $\alpha$ and $\beta_3$. For each setting, we train 8 models using $\alpha \in \{1, 5, 10, 15, 20\}$ and $\beta_3 \in \{0.999, 0.9999\}$. All of the AdEMAMix models are shown in Fig. 6 and Fig. 31. Only the best AdamW model is shown in Fig. 31.

Table 6: **Hyperparameters tuning for our ViT AdamW experiments.** We started our hyperparameter search from the values given in (Dosovitskiy et al. (2021b), Table 3), which recommends using (learning-rate, weight decay, dropout)= $(0.003, 0.3, 0.1)$ when training on ImageNet-1k and $(0.001, 0.03, 0.1)$ when training on ImageNet-21k. The various hyperparameters we experimented with are in the following table, values which gave us the lowest validation loss are in bold. For each setting, AdEMAMix experiments use the same hyperparameters as the best AdamW baselines.

| Setting | 24M params, 11M images (ImageNet-21k) | 86M params, 11M images (ImageNet-21k) | 86M params, 1.3M images (ImageNet-1k) |
|---|---|---|---|
| Learning rate | $0.001, \mathbf{0.003}, 0.005$ | $\mathbf{0.001}, 0.003$ | $0.001, 0.003, \mathbf{0.005}$ |
| Weight decay | $\mathbf{0.03}, 0.1, 0.3$ | $0.03, \mathbf{0.1}, 0.3$ | $0.03, 0.1, \mathbf{0.3}, 0.5, 0.7$ |
| Dropout | $0.1$ | $0.1$ | $0.1$ |
| AdamW $\beta_1$ | $\mathbf{0.9}, 0.99, 0.999$ | $\mathbf{0.9}, 0.99, 0.999$ | $\mathbf{0.9}, 0.99, 0.999$ |
| Batch size | $4096$ | $4096$ | $4096$ |

## C ADDITIONAL RESULTS

### C.1 LLM EXPERIMENTS

#### C.1.1 IN-CONTEXT LEARNING (ICL) RESULTS

**In-Context Learning (ICL) results.** We evaluate our largest (1.3B) LLM models on in-context learning tasks. We use the `lm-eval` package (Gao et al., 2023). We consider the following tasks:

HellaSwag (Zellers et al., 2019), Winogrande (Sakaguchi et al., 2020), ARC (Bhakthavatsalam et al., 2021), BoolQ (Clark et al., 2019), LogiQA (Liu et al., 2020), MathQA (Amini et al., 2019), MMLU (Son et al., 2024), OpenbookQA (Mihaylov et al., 2018), PIQA (Bisk et al., 2020), PubmedQA (Jin et al., 2019), RewardBench (Lambert et al., 2024), Sciq (Welbl et al., 2017), TruthfulQA (Lin et al., 2022).

The two AdEMAMis and AdamW models we benchmark have been trained for 1M steps, with a batch size of 128, corresponding to around 131B tokens. The results of the evaluation are in Table 7. We observe how the model trained with AdEMAMix is outperforming the AdamW model on nearly all of the tasks, sometimes by a large margin for e.g. PubmedQA. In addition to Table 7, we track the evolution of some of those scores during training for MMLU, RewardBench, ARC-Challenge, and ARC-Easy, results can be seen in Fig. 8. For that figure, we modify the way MMLU is evaluated. Instead of appending all the multiple answers, we concatenate the question prompt and each answer separately, and pick the most likely answer. We found it allows a much better comparison of smaller models, which otherwise fluctuates around random guessing.

Table 7: **In-context learning results for our** 1.3**B parameter models.** We compare AdamW and AdEMAMix models trained on 131B tokens. We use the `lm-eval` package.

| Task | AdamW | AdEMAMix | Task | AdamW | AdEMAMix |
|------|-------|----------|------|-------|----------|
| ARC-Challenge | 0.262 | **0.274** | OpenbookQA | **0.240** | 0.238 |
| ARC-Easy | 0.612 | **0.619** | PIQA | **0.715** | **0.715** |
| BoolQ | 0.569 | **0.576** | PubmedQA | 0.556 | **0.632** |
| HellaSwag | 0.426 | **0.436** | RewardBench | 0.569 | **0.573** |
| LogiQA | **0.235** | 0.225 | RewardBench (reasoning) | 0.617 | **0.630** |
| MathQA | 0.226 | **0.236** | Sciq | 0.903 | **0.907** |
| Winogrande | 0.563 | **0.580** | TruthfulQA | **0.361** | 0.352 |
| MMLU | 0.244 | **0.248** | | | |

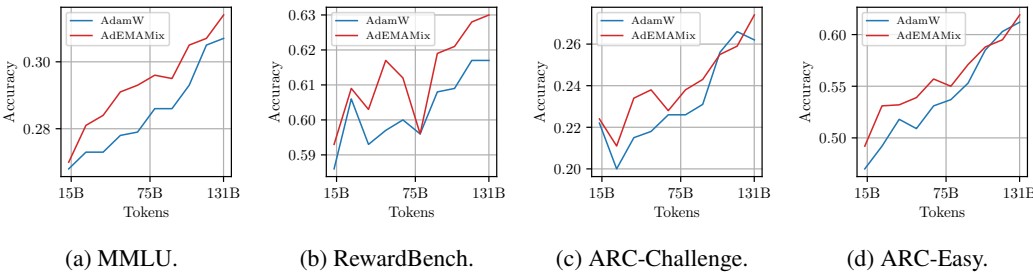

(a) MMLU.    (b) RewardBench.    (c) ARC-Challenge.    (d) ARC-Easy.

Figure 8: **In-context learning results for our** 1.3**B models.** We periodically measure the performances of AdamW and AdEMAMix models on in-context learning tasks. Except for a couple exceptions, AdEMAMix is performing better than AdamW.

### C.1.2 RESULTS USING 3B PARAMETER MODELS

**Experimental setup.** We train 3B parameter models on RedPajama v2 data. We use a learning rate of $\eta = 0.0003$, a weight decay $\lambda = 0.1$, a gradient clipping value of $0.1$, a batch size of $1024$. The sequence length is $1024$. We use a cosine $\eta$ decay and $3000$ steps of warmup. For AdEMAMix we used $\beta_3 \in \{0.9999, 0.99995\}$ and $\beta_1 \in \{0.0, 0.9\}$. For AdamW, we use $\beta_1 = 0.9$. Both methods use $\beta_2 = 0.999$. We train models for $477k$ iterations, on a total of $500$B tokens.

**Results.** Results are shown in Fig. 9. We observe that AdEMAMix is outperforming the AdamW baseline. Moreover, using $\beta_1 = 0.0$ in AdEMAMix is not deteriorating the results. More experiments are required to estimate how many additional steps would be needed for the AdamW baseline to reach the same performances as AdEMAMix.

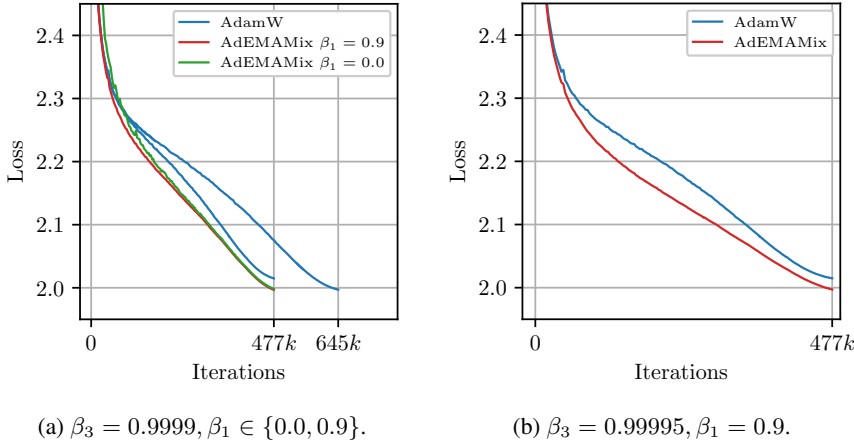

(a) $\beta_3 = 0.9999, \beta_1 \in \{0.0, 0.9\}$.   (b) $\beta_3 = 0.99995, \beta_1 = 0.9$.

Figure 9: **Results using 3B parameter models.** For AdamW models, we use $\beta_1 = 0.9, \beta_2 = 0.999$. In **(a)** we compare the AdamW baseline to AdEMAMix models trained with $\beta_3 = 0.9999$ and $\beta_1 \in \{0.0, 0.9\}$. Both AdEMAMix models reach a similar final loss and outperform the baseline. The convergence to the solution is slower at the beginning for $\beta_1 = 0.0$. When $\beta_1 = 0.0$, the memory footprint of AdEMAMix is the same as for AdamW, meaning that we can get the improvement in loss at no extra cost. In **(b)**, we train an AdEMAMix model with a larger $\beta_3 = 0.99995$. While we observe a lower loss in the early training phase, the final loss is similar to the one obtained with $\beta_3 = 0.9999$. AdEMAMix reaches the same loss as an AdamW model trained for $168k$ more steps.

### C.1.3 MORE RESULTS ON FORGETTING

**Evolution of forgetting during training.** In Fig. 4 in the main paper, we follow the loss on one specific batch over the entire training. In the following experiment, we do a closeup on the forgetting of different batches as training progresses. The goal being to visualize how later batches are ultimately more remembered, and compare the forgetting profiles of AdamW and AdEMAMix. For this, in Fig. 10a and Fig. 10b, we follow fixed batches at different stages of the training process. Every $10k$ iterations, we measure the loss over a specific batch $B$ before and after training on that batch.

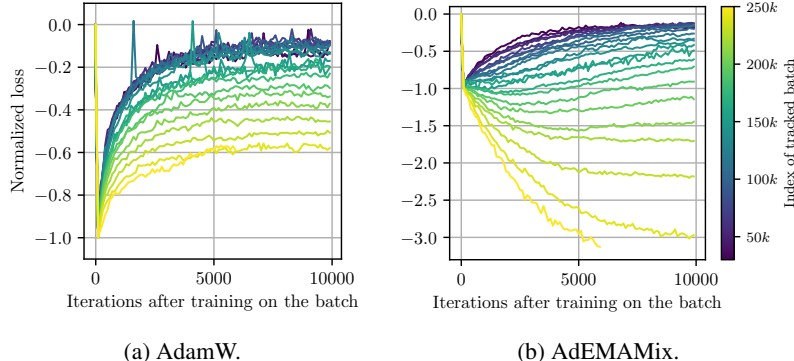

(a) AdamW.                          (b) AdEMAMix.

Figure 10: **Comparing forgetting between AdamW and AdEMAMix.** In **(a)** and **(b)**, we follow—over $10k$ iterations—the loss of training batches $B_\tau$ for $\tau \in \{30k, 40k, \dots, 250k\}$. We measure the loss right before training on $B_\tau$, and then monitor this loss over the next $10k$ steps. For instance, the most yellow curve represents the loss on batch $B_{250k}$ that we track for the last $6k$ iterations of training (we train for $256k$ iterations). The loss is normalized such that the loss for $B_\tau$ is $0$ and the one at time $\tau + 50$ is $-1$. This allows us to overlap curves and observe the tendency of the model to forget batches throughout training. We observe that AdEMAMix improves its loss on training batches $B_\tau$ over many timesteps, a striking difference compared to AdamW, which has its loss on $B_\tau$ increasing faster. Moreover—likely due to the learning rate scheduler—the models forget significantly slower at the end of training. In this later training stage ($\tau > 180k$), the difference between the two optimizers is especially pronounced. While this visualization allows to efficiently compare forgetting at different stages of training, the normalization prevents us from comparing the drop in loss between AdEMAMix and AdamW. To compare loss values, see Fig. 4.

**Why are training batches forgotten more at the end of training?** Given the previous observation about how both AdamW and AdEMAMix models are forgetting the training batches at different paces given different stages of training, one can wonder what is causing this phenomenon? To answer this question, we contrast the results of Fig. 4 with results obtained using a different scheduler with a constant learning rate and a linear decay. We train 110M parameter models for $300k$ iterations, using a max learning rate of $0.001$ and a batch size of $64$. Results for those experiments are in Fig. 11. Those results indicate that the decaying learning rate might be the main parameter controlling how much a batch is remembered during training. This has interesting implications when selecting which learning rate decay to use. A cosine decay, with a rather long period of decay, might remember more than a constant learning rate scheduler with only a small number of steps of linear decay at the end.

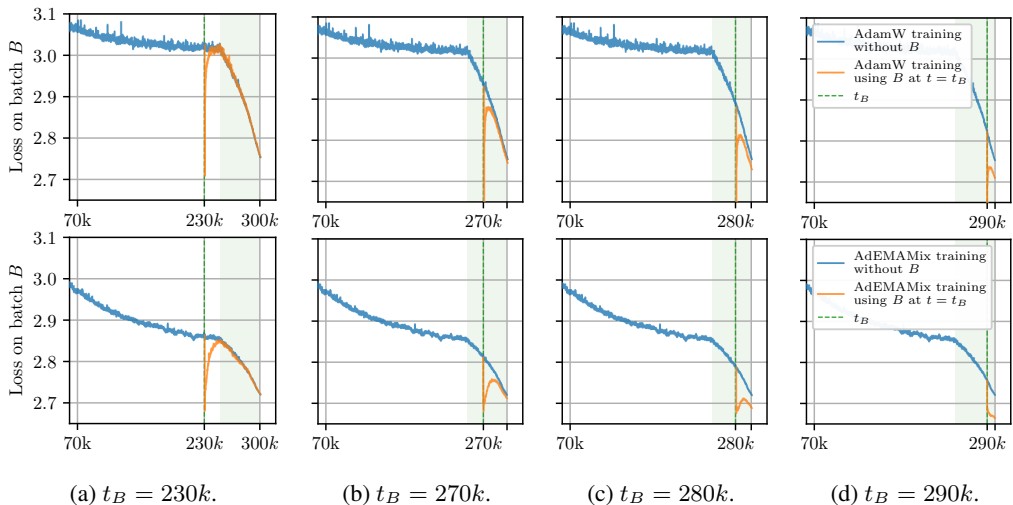

(a) $t_B = 230k$.      (b) $t_B = 270k$.      (c) $t_B = 280k$.      (d) $t_B = 290k$.

Figure 11: **Measuring forgetting using a held-out batch $B$, using a constant $\eta$-scheduler with linear decay.** The top row is for AdamW, the bottom row is for AdEMAMix. The setup is similar to the one from Fig. 4, except that we now use a different learning rate scheduler. We use 3000 steps of linear warmup, them $247k$ steps of constant learning rate $\eta = 0.001$, and finally $50k$ steps of linear decay (represented by the shaded green area in the above plots). In Fig. 4, we observed that the batches used later during training are remembered more by both AdamW and AdEMAMix. Multiple hypotheses could explain this phenomenon: (i) it could be that the learning rate decay is the main parameter controlling forgetting, or (ii) it could be that a sufficient number of steps are required to reach a part of the optimization landscape where gradients are more orthogonal and baches are remembered. The present experiment refutes that second hypothesis. Indeed, before the start of the $\eta$-decay (as seen in **(a)**), training batches are not remembered at the end of training. They only start to be remembered when the decay periods starts, as seen in **(b,c,d)**. Moreover, we can make the same observation as in Fig. 4: compared to AdamW, AdEMAMix models tend to keep the training data longer in memory.

### C.1.4 REMOVING THE SECOND EMA MID-TRAINING

**Removing the second EMA mid-training.** In Fig. 5b and Fig. 5c, we looked at what is happening when we switch from AdamW to AdEMAMix in the middle of training. In this section, we will look at the opposite conversion: removing the $m_2$ parameter of AdEMAMix during training, effectively switching back to AdamW. Results for this experiment can be seen in Fig. 12. Right after the switch, we observe a drop of the loss, followed by an increase and finally back to convergence. Ultimately, the final loss value is in between the ones obtained training only using AdamW and only using AdEMAMix.

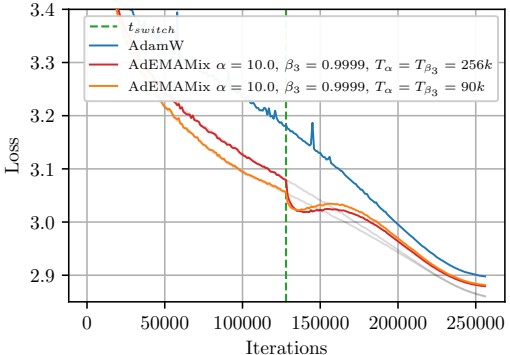

Figure 12: **Switching from AdEMAMix to AdamW at** $t_{switch} = 128k$**.** Using our AdEMAMix optimizer, we train two 110M parameters models with different $T_\alpha = T_{\beta_3} \in \{90k, 256k\}$. For historical reasons, those experiments were done using a batch size of $32$ instead of $64$ (as we used for all the other 110M parameter experiments in this paper). We switch from AdEMAMix to AdamW at $t_{switch} = 128k$. At first, the removal of the $+\alpha m_2$ contribution in the update is suddenly decreasing the effective learning rate, causing the loss to drop suddenly. The loss then increases slightly and ends up in between the AdamW only (blue curve) and AdEMAMix only (grey curves) losses.

### C.1.5 HYPERPARAMETER SENSITIVITY

**Hyperparameter sensitivity.** Depending on whether the $\alpha$ and $\beta_3$ schedulers are used, AdEMAMix adds up to 4 new hyperparameters: $\alpha$, $\beta_3$, $T_\alpha$ and $T_{\beta_3}$. In all our experiments we tied $T_\alpha = T_{\beta_3} = T_{\alpha,\beta_3}$. In this section we analyze the sensitivity of AdEMAMix to those hyperparameters. We study the impact of $\alpha$ and $\beta_3$ in Fig. 13, revealing wide ranges of values for which hyperparameters are outperforming the AdamW baseline. We study the sensitivity of $T_{\alpha,\beta_3}$ as well as justify the choice of using a scheduler on $\alpha$ in Fig. 14. When training from scratch $T_{\alpha,\beta_3}$ needs simply to be large enough to avoid early divergence. In Fig. 16 we study the sensitivity to the gradient clipping and AdEMAMix's $\beta_1$ value. While gradient clipping can help stabilize training and smooth the loss curves, it has little impact over the final loss value. Reducing the value of $\beta_1$, we observe some loss spikes, yet the final loss value is relatively unaffected.

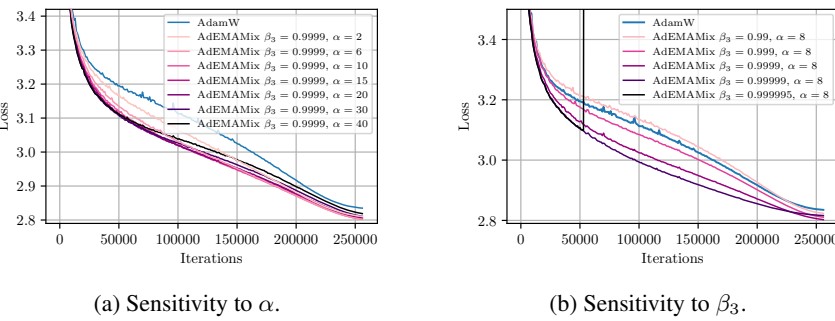

(a) Sensitivity to $\alpha$.  (b) Sensitivity to $\beta_3$.

Figure 13: **Sensitivity of AdEMAMix to $\alpha$ and $\beta_3$.** We test the sensitivity of 110M parameter models to the values of $\alpha$ and $\beta_3$. In **(a)**, we vary $\alpha \in \{2, 6, 10, 15, 20, 30, 40\}$ while keeping all other parameters equal ($\beta_3 = 0.9999, T_{\alpha,\beta_3} = 256k, \text{clipping} = 0.5, \eta = 0.001, \lambda = 0.1$). We see that the larger the $\alpha$, the more significant is the loss decrease early during training, yet this does not necessarily translates into a better final loss. The last iterate's loss is larger for large values of $\alpha$. For this model, a sweet spot seems to be reach for $\alpha = 10$. Moreover, the span of values resulting in good results is very large. In **(b)**, we now vary $\beta_3 \in \{0.99, 0.999, 0.9999, 0.99999, 0.999995\}$. We first observe that—except for $\beta_3 = 0.999995$—all those values are outperforming the AdamW baseline. Moreover, we observe that there is a sweet spot, at $\beta_3 = 0.9999$, after which the final loss increases.

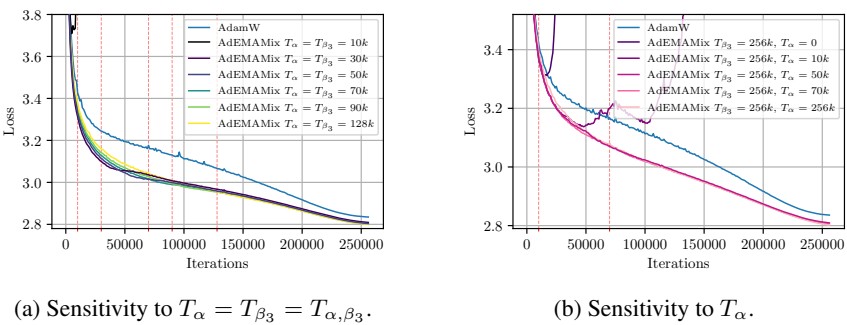

(a) Sensitivity to $T_\alpha = T_{\beta_3} = T_{\alpha,\beta_3}$.  (b) Sensitivity to $T_\alpha$.

Figure 14: **Sensitivity of AdEMAMix to $T_\alpha$ and $T_{\beta_3}$.** For all the AdEMAMix experiments in this figure, we used $\beta_1 = 0.9, \beta_2 = 0.999, \beta_3 = 0.9999, \alpha = 10$. In **(a)**, we test the sensitivity of 110M parameter models to the values of $T_\alpha$ and $T_{\beta_3}$. We tied the two values as in all the experiments in the main paper: $T_\alpha = T_{\beta_3} = T_{\alpha,\beta_3}$. We observe that when not enough warmup steps are used, the iterates either diverge or converge to slightly worse loss values. In **(b)** we set $T_{\beta_3} = 256k$ and only vary $T_\alpha$. We observe that $T_\alpha$ needs to be large enough to avoid divergence. This demonstrates the necessity of using a warmup on $\alpha$. This being said, the necessity of increasing $\alpha$ linearly depends on the value of $\alpha$. Using a smaller $\alpha$ value can work with $T_\alpha = 0$, but using an $\alpha$-scheduler alleviates any instability issue and increases the range of optimal $\alpha$ values (see Fig. 15). Moreover, in other experiments, we observe how not using schedulers yields detrimental large updates in the early training phase, see App. C.1.10, Fig. 28.

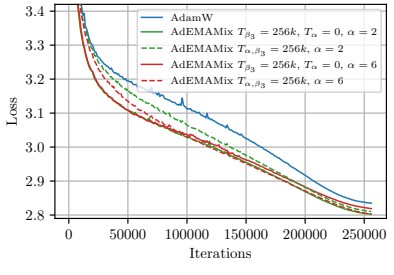 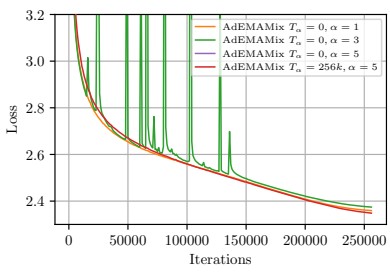

(a) No $\alpha$-scheduler (110M).    (b) No $\alpha$-scheduler (1.3B).

Figure 15: **The $\alpha$-scheduler reduces the sensitivity to $\alpha$.** We compare AdEMAMix experiments with and without $\alpha$-scheduler. The experiments with $\alpha$-scheduler are more stable. In **(b)**, for 1.3B parameter models, only $\alpha = 1$ converges in a stable fashion (the purple curve with $\alpha = 5$ diverged early), and reaches a final loss slightly worse than the AdEMAMix model trained using an $\alpha$-scheduler over $256k$ steps. In **(a)**, for smaller 110M parameter models, we observe (i) that it is possible to converge without $\alpha$-scheduler for small $\alpha$ values, as well as (ii) that it seems easier to reach good loss values with a scheduler (the dotted lines have a lower final loss on average).

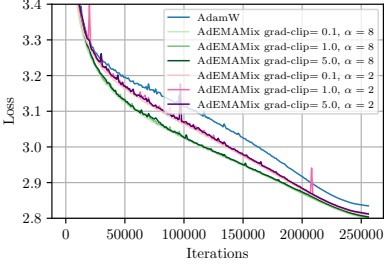 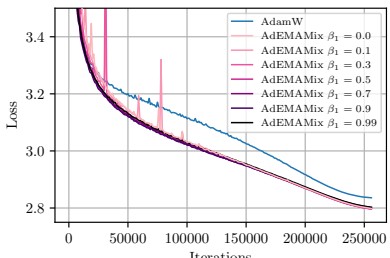

(a) Sensitivity to gradient clipping.    (b) Sensitivity to $\beta_1$.

Figure 16: **Sensitivity of AdEMAMix to $\beta_1$ and the gradient clipping value.** Unless specified in the caption, all the AdEMAMix experiments in this figure used $\beta_1 = 0.9, \beta_2 = 0.999, \beta_3 = 0.9999, \alpha = 10, T_{\alpha,\beta_3} = 256k$. In **(a)** we vary the amount of gradient clipping used during training. We notice how gradient clipping can help to smooth the curves, yet bringing only minor improvements. In **(b)** we perform a sweep over $\beta_1 \in \{0, 0.1, 0.3, 0.5, 0.7, 0.9, 0.99\}$. The value $\beta_1 = 0.0$ is especially interesting as it allows us to remove entirely $\boldsymbol{m}_1$, therefore saving memory. We observe that smaller $\beta_1$ values yield noisier curves, with multiple loss spikes. At the 110M parameter scale, those spike do not significantly impact the convergence, we conjecture that they could become a problem at larger scales. We also notice that slightly better losses could be obtained using smaller $\beta_1$ values (e.g. $\beta_1 = 0.3$), we keep $\beta_1 = 0.9$ in all of our experiments for the ease of comparison with AdamW.

### C.1.6 IMPACT OF TRAINING FOR FEWER ITERATIONS

**Sensitivity to the number of iterations.** As we rely on old gradients, on question that arises is whether AdEMAMix would still perform well when reducing the number of iterations. In this section we compare the loss obtained by 110M parameter models when halving the number of iterations and doubling the batch size, in such a way that the number of tokens consumed for training is always the same (17B). In Fig. 17, we observe that decreasing the number of iterations too much increases the final loss of the model. This effect is more pronounced for AdEMAMix. However, at both $32k$ and $64k$ iterations, AdEMAMix still outperforms AdamW. In Fig. 18 we observe that reducing $\beta_3$ can mostly alleviate the problem.

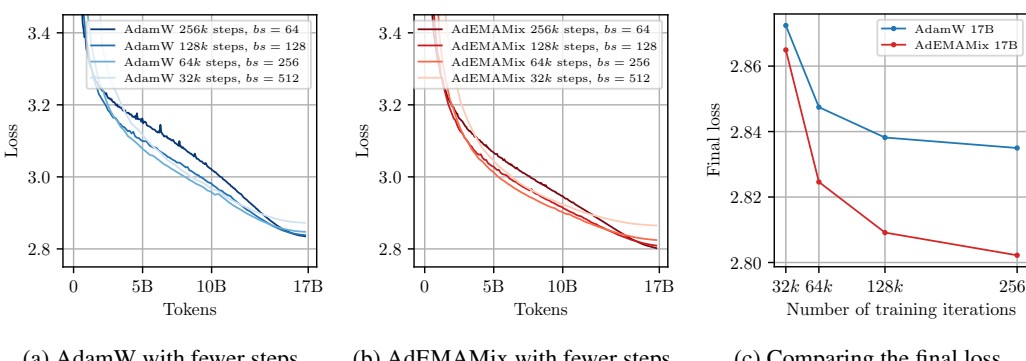

(a) AdamW with fewer steps.    (b) AdEMAMix with fewer steps.    (c) Comparing the final loss.

Figure 17: **Impact of using fewer iterations on AdEMAMix vs. AdamW.** We train multiple 110M parameter models. All models are trained on 17B tokens, but using different batch sizes ($bs \in \{64, 128, 256, 512\}$), which results in different numbers of iterations: $\{32k, 64k, 128k, 256k\}$. For both AdamW and AdEMAMix, we use hyperparameters from Table 2 ($\beta_3 = 0.9999$ for AdEMAMix). In **(a)**, we observe a drop in performance when reducing the number of steps, as the loss of the final iterates is higher for the model trained for $32k$ steps compared to the ones trained for more iterations. In **(b)** we observe a similar story for AdEMAMix models, with a slightly more pronounced performance degradation. Yet, as it can be seen in **(c)**, the gap between AdEMAMix and AdamW still remains, even when training for $32k$ iterations. Importantly, this experiment does not suggest that AdEMAMix is performing worse when using larger batches. In our ViT experiments, we successfully use a batch size of $4096$.

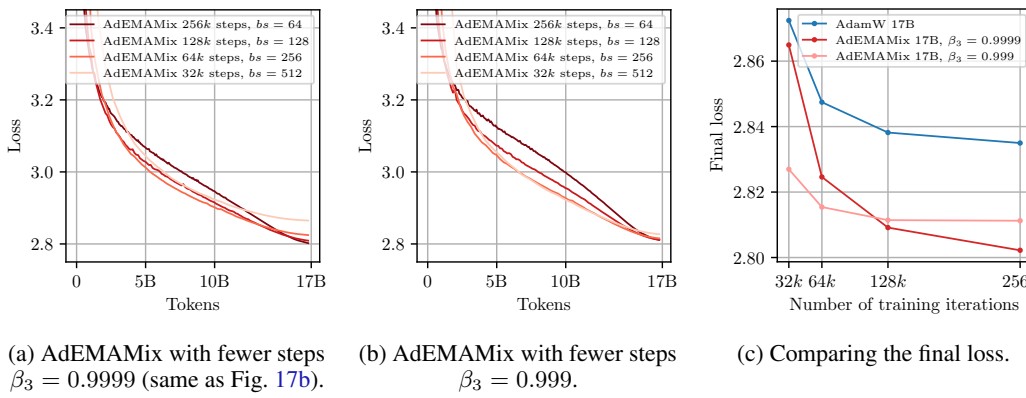

(a) AdEMAMix with fewer steps $\beta_3 = 0.9999$ (same as Fig. 17b).    (b) AdEMAMix with fewer steps $\beta_3 = 0.999$.    (c) Comparing the final loss.

Figure 18: **Reducing $\beta_3$ can work better with low numbers of iterations.** We take the same experimental protocol as Fig. 17 but this time we compare training AdEMAMix models with $\beta = 0.999$ and $\beta = 0.9999$. In **(a,b)** we observe how using a smaller $\beta_3 = 0.999$ solves—in a large part—the problem of decreasing performances when using a small number of iterations. In **(c)**, while final loss for $T \in \{128k, 256k\}$ is lower using $\beta_3 = 0.9999$, using $\beta_3 = 0.999$ is better for $T \in \{32k, 64k\}$. Using $\beta_3 = 0.999$, the gap between AdEMAMix and AdamW increases as the number of iterations decreases.

C.1.7 LIMITATIONS OF A SINGLE EMA

**Results on a 2D toy example.** A natural question that arises from our method is whether it is possible to obtain the same results without the additional EMA. In this section we aim to convince the reader that the two EMAs are required. First, we propose to study a small toy 2D optimization problem:

$$(x^\star, y^\star) \in \arg\min_{(x,y)} f(x,y), \quad \text{with } f(x,y) = 8(x-1)^2 \times (1.3x^2 + 2x + 1) + 0.5(y-4)^2$$

This function was introduced by Liu et al. (2023) as a function with sharp curvature along the $x$-axis, and flatter curvature along the $y$-axis. Initializing the first iterate $\boldsymbol{x}^{(0)} = (0.3, 1.5)$, we start optimizing with (i) Adam with a large $\beta_1 = 0.999$, and (ii) AdEMAMix with $\beta_1 = 0.9$ and $\beta_3 = 0.999$. In both cases $\beta_2 = 0.999$. To make the experiment interesting, we initialize the EMA buffers for both methods to either $(-3, 0)$ or $(-0.8, -3)$. This has for effect to give an initial "speed" to the first iterate. As a result of this speed, Adam with a large $\beta_1$ is unable to correct his trajectory. In contrast, using two EMAs, one with a small, and one with a large $\beta$—as in AdEMAMix—converges to the solution. The fast changing EMA can correct for the bias introduced by the slow changing EMA. See Fig. 19 for results.

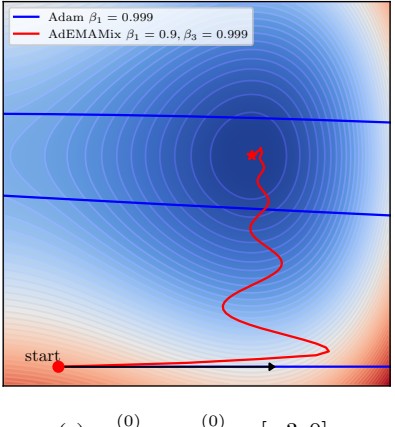
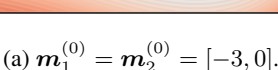
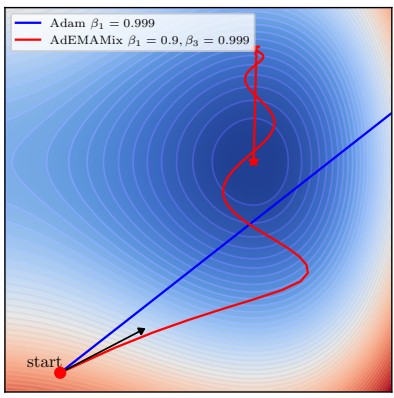

(a) $\boldsymbol{m}_1^{(0)} = \boldsymbol{m}_2^{(0)} = [-3, 0]$.          (b) $\boldsymbol{m}_1^{(0)} = \boldsymbol{m}_2^{(0)} = [-0.8, -3]$.

Figure 19: **Showing the necessity of a second momentum term on a toy example.** In this experiment, we use Adam and AdEMAMix to optimize a simple function (same function as in Fig.2 in Liu et al. (2023), $\star$ is a local optimum). We want to understand what is preventing the use of large $\beta_1$ values in Adam, and how AdEMAMix is overcoming this problem. Starting from $\boldsymbol{x}^{(0)} = [0.3, -1.5]$, we initialize $\boldsymbol{m}_1$ (resp. $\boldsymbol{m}_1$ and $\boldsymbol{m}_2$) for Adam (resp. AdEMAMix) to either $[-3, 0]$ or $[-0.8, -3]$—shown with black arrows (not to scale). $\boldsymbol{\nu}^{(0)}$ is still initialized to $\boldsymbol{0}$. This is similar to giving an initial speed to the iterate. In both cases **(a)** and **(b)**, the iterates following the Adam trajectory fail to converge in the given number of iterations. This is due to the large $\beta_1$ which requires many hundreds of iterations to significantly alter $\boldsymbol{m}_1$. As a result, the iterates mostly follow the initial direction imposed by $\boldsymbol{m}_1^{(0)}$ ($\boldsymbol{\nu}$ is still used to scale $\boldsymbol{m}_1$). In contrast, AdEMAMix converges to the solution. The fast (smaller $\beta$) EMA of AdEMAMix corrects the trajectory of the slow (high $\beta$) EMA. Interestingly, AdEMAMix initially overshoots the optimum. It has to wait for the influence of $\boldsymbol{m}_2^{(0)}$ to fade away before finally converging to the solution. While this experiment does not explain why a larger momentum would be beneficial, it illustrates how using two EMAs can enable the use of large $\beta$ values. A better intuition behind why a large momentum could be beneficial is presented in Fig. 2.

**Results training** 110**M parameters LMs.** To further demonstrate that simply increasing $\beta_1$ in AdamW does not provide nearly the same gains as AdEMAMix, we run several additional experiments. In Fig. 20 we show what we obtain when training from scratch using a single EMA with a $\beta_1 \in \{0.99, 0.999, 0.9999, 0.99999\}$. Naively increasing $\beta_1$ in AdamW does not work; adding a scheduler on $\beta_1$ to smoothly increase its value during the entirety of the training also fails. In Fig. 21, we show results when increasing $\beta_1$ in the middle of training, with and without scheduler on $\beta_1$. This differs from the previous setting as we bypass the initial training phase capable of causing instabilities (see

§ 3), as well as bypass the initial iterations during which the bias correction done by AdamW can have an impact. Here again we observe increasing the $\beta_1$ value does not provide any noticeable gain.

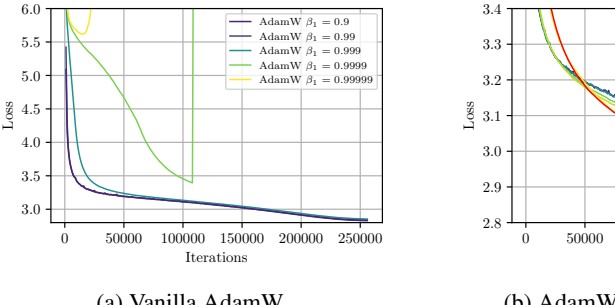

(a) Vanilla AdamW.  (b) AdamW with $\beta_1$-scheduler.

Figure 20: **Increasing $\beta_1$ in AdamW.** In **(a)** we perform a simple sweep over the $\beta_1$ hyperparameter of AdamW, we observe how the optimizer is unable to cope with large values of $\beta_1$, diverging for $\beta_1 > 0.999$. In **(b)** we modify AdamW to incorporate the tricks we used in AdEMAMix, i.e. we add a scheduler on $\beta_1$. While this prevents divergence, the final loss is getting worse as we increase $\beta_1$. We had to increase $\beta_2$ to stabilise the training when using $\beta_1 = 0.99999$. Those experiments support the importance of the additional EMA in AdEMAMix.

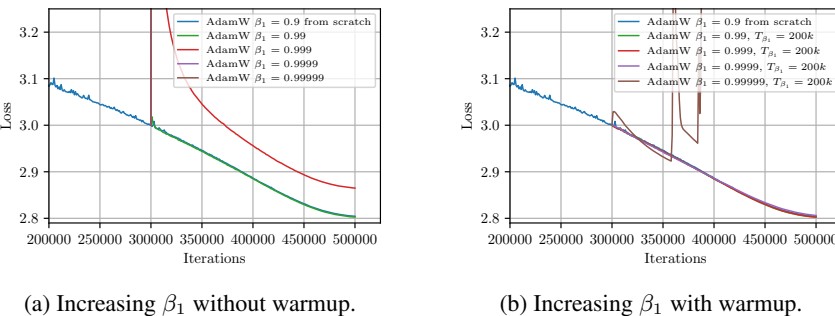

(a) Increasing $\beta_1$ without warmup.  (b) Increasing $\beta_1$ with warmup.

Figure 21: **Increasing $\beta_1$ in AdamW later during training.** In **(a)** we load an AdamW checkpoint and resume training using AdamW with a larger $\beta_1$. We observe no gain over the baseline from increasing the value of $\beta_1$. In **(b)**, we add a scheduler over $\beta_1$ (similar to the scheduler over $\beta_3$ for AdEMAMix) to smooth the transition between $\beta_1 = 0.9$ and larger values. We set $T_{\beta_1} = 200k$ iterations. Despite the smoother transition, no gain over the baseline is observed.

C.1.8    REMOVING $m_1$ BY SETTING $\beta_1 = 0$

**Effect of the batch size when $\beta_1 = 0$.** In this section we investigate the impact of setting $\beta_1 = 0$, which allows to replace $\boldsymbol{m}_1^{(t)}$ by the current gradient $\boldsymbol{g}^{(t)}$, saving memory. In this scenario the memory complexity of AdEMAMix is the same as the one of AdamW. We noticed in Fig. 16b how reducing $\beta_1$ can make the training slightly more unstable, triggering some loss spikes. In Fig. 22 we study the impact of the batch size on those instability, revealing that $\beta_1 = 0$ is less stable when using larger batch sizes. However, despite the spikes, the final loss for $\beta_1 = 0$ and $\beta_1 = 0.9$ are very similar—$\beta_1 = 0.9$ yielding slightly better results for smaller batch sizes.

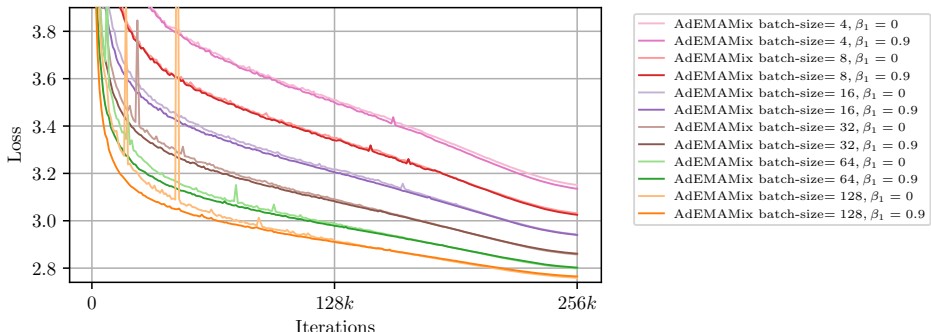

Figure 22: **Effect of batch size with $\beta_1 = 0$.** For a 110M parameter model, we use $\beta_1 \in \{0.0, 0.9\}$ and vary the batch-size $\in \{4, 8, 16, 32, 64, 128\}$. Lighter curves use $\beta_1 = 0$, darker curves are their $\beta_1 = 0.9$ counterpart. For small batch sizes, no instabilities are observed, and $\beta_1 = 0.9$ reaches slightly better performances. For larger batch sizes, instabilities are observed with using $\beta_1 = 0$, yet the final loss values are similar as the ones obtained with $\beta_1 = 0.9$. This indicates that one can potentially save memory without compromising results.

**Setting $\beta_1 = 0$ for 1.3B parameter models.** In this section, using a 1.3B parameter model, we re-do the same experiment as in Fig. 1, but we set $\beta_1 = 0$. In Fig. 23, we show the results of this experiment.

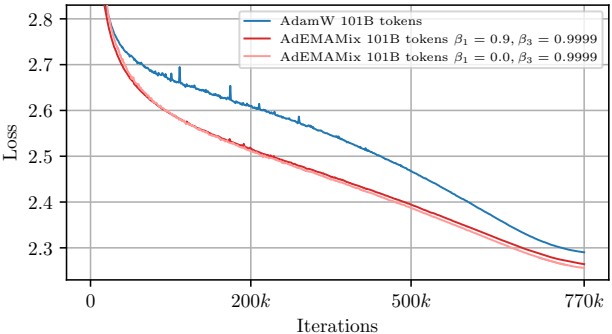

Figure 23: **Using $\beta_1 = 0$ with a 1.3B parameter model.** We use $\beta_2 = 0.999$ for all those experiments. We observe how it is possible to converge despite using $\beta_1 = 0$. The final loss with $\beta_1 = 0$ is slightly better than the one with $\beta_1 = 0.9$.

C.1.9   COMPARING $m_1 + \alpha m_2$ VERSUS $(1 - \alpha)m_1 + \alpha m_2$

In the equation AdEMAMix, we combine $m_1$ and $m_2$ using $m_1 + \alpha m_2$. In this section we contrast this to using a convex combination $(1 - \alpha)m_1 + \alpha m_2$, with $\alpha \in [0, 1]$.

**Are those two somewhat equivalent?** Given the combination $\eta(m_1 + \alpha m_2)$, it is possible to rewrite it as a convex combination as such:

$$\eta(m_1 + \alpha m_2) = \hat{\eta}((1 - \hat{\alpha})m_1 + \hat{\alpha}m_2), \quad \text{with } \hat{\eta} = \eta(\alpha + 1) \text{ and } \hat{\alpha} = \frac{\alpha}{\alpha + 1}.$$

Therefore, the two formulas are equivalent up-to a reparametrization. However, in practice, we use schedulers on $\eta$ and $\alpha$. While at any timestep $t$, it is possible to reparametrize one formula into the other, there is not one single reparametrization that holds for all timesteps. Assuming that we do not change the nature of the schedulers for $\alpha$ and $\eta$, those two formulations are therefore *not equivalent*. In Fig. 24 we show the evolution of the weights given to $m_1$ and $m_2$ for the two approaches. We assume a cosine scheduler for $\eta$ and a linear scheduler for $\alpha$. We can see how the two formulations give very different weights to $m_1$ and $m_2$.

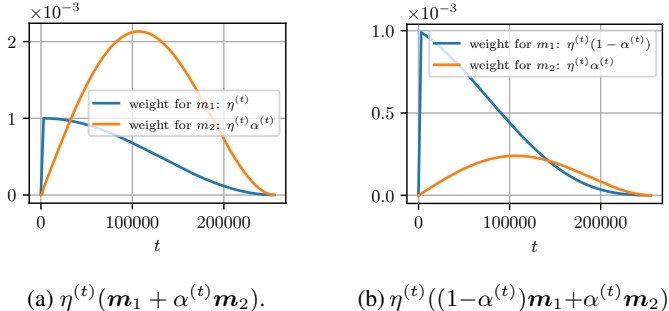

(a) $\eta^{(t)}(m_1 + \alpha^{(t)}m_2)$.          (b) $\eta^{(t)}((1-\alpha^{(t)})m_1+\alpha^{(t)}m_2)$.

Figure 24: **Weights given to $m_1$ and $m_2$ for the two formulations.** In **(a)**, we show the weights given to $m_1$ and $m_2$ when we use $\eta^{(t)}(m_1 + \alpha^{(t)}m_2)$. We use $\alpha = 8$ ($\alpha$ is the final value reached by the scheduler $\alpha^{(t)}$). In **(b)**, we show the weights given to $m_1$ and $m_2$ when we use $\eta^{(t)}((1 - \alpha^{(t)})m_1 + \alpha^{(t)}m_2)$. We use $\alpha = 0.9$. Importantly, in both cases, we assume a cosine scheduler for $\eta^{(t)}$ and a linear scheduler for $\alpha^{(t)}$. Given this choice of schedulers, there is no reparametrization which can make those two formulation equal. In Fig. 25 we show that using the convex combination formulation yields worse results.

**Is the convex combination better?** Now that we established that the two formulations are different, which one is better? To answer this we run two grid searches, varying $\eta \in \{0.4, 0.6, 0.8, 1, 1.2, 1.4, 1.6\}$ and $\alpha \in \{2, 4, 6, 8, 10, 12, 14\}$ for the formulation in equation AdEMAMix, and $\alpha \in \{0.3, 0.4, 0.5, 0.6, 0.7, 0.8, 0.9\}$ for the convex combination formulation. Results in Fig. 25 show how the formulation used in AdEMAMix provides a better loss *for all the pairs* $(\eta, \alpha)$ *tested*, also outperforming AdamW for all of those. In contrast, for the convex combination, while most $(\eta, \alpha)$ pairs outperform AdamW, the improvement is smaller.

**Getting the same results by changing the schedulers for the convex combination.** We concluded before that—unless we tamper with the schedulers—it is not possible to find a reparametrization that would work for all timesteps and make the two formulations equivalent. What if we could change the schedulers? Which schedulers would we need to use with the convex formulation, to get the same results as with the other formulations. The answer is simple, let $\eta^{(t)}$ and $\alpha^{(t)}$ denote the cosine and linear schedulers for resp. $\eta$ and $\alpha$. The convex combination would need to use the following schedulers:

$$\hat{\eta}^{(t)} = \eta^{(t)}(\alpha^{(t)} + 1), \qquad \hat{\alpha}^{(t)} = \frac{\alpha^{(t)}}{\alpha^{(t)} + 1}.$$

Therefore, to get equivalent results as for the original AdEMAMix formulation, the learning rate scheduler would need to be $\hat{\eta}^{(t)}$, which is not a cosine scheduler. The $\alpha$ scheduler, $\hat{\alpha}^{(t)}$, would not be linear. We plot those schedulers in Fig. 26.

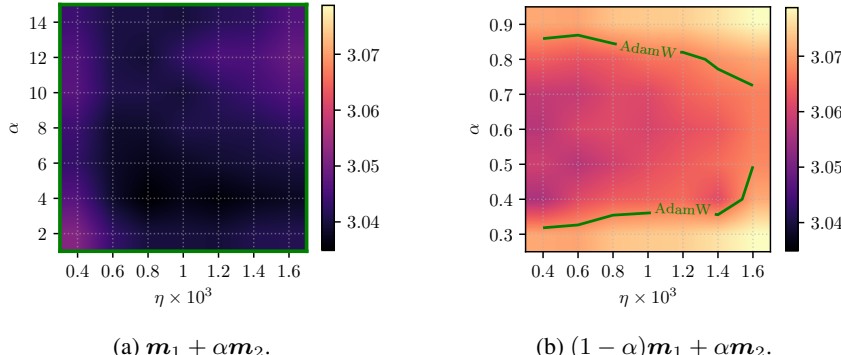

(a) $\boldsymbol{m}_1 + \alpha \boldsymbol{m}_2$.

(b) $(1 - \alpha)\boldsymbol{m}_1 + \alpha \boldsymbol{m}_2$.

Figure 25: $\eta$-$\alpha$ **hyperparameter sweep for the two formulations.** For pairs of $(\eta, \alpha)$, we train small (6-layers) transformer LMs on RedPajama v2 data. We plot the results for the two formulations, the color represents the final validation loss, and is normalized between the two plots. We also do a sweep over $\eta$ to fine the value giving the best results for AdamW. We use a green level curve to show the performance of the best AdamW model. Points in the interior of the green level curve represents configurations outperforming AdamW. In **(a)**, we color the entire frame in green as all the pairs of hyperparameters are outperforming AdamW. It is clear from those two figures that using $\boldsymbol{m}_1 + \alpha \boldsymbol{m}_2$ yields better results, more consistently, compared to using $(1 - \alpha)\boldsymbol{m}_1 + \alpha \boldsymbol{m}_2$.

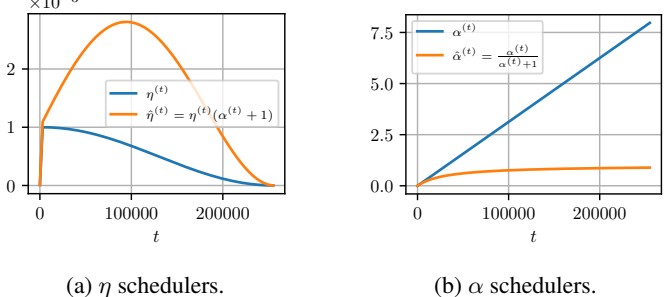

(a) $\eta$ schedulers.

(b) $\alpha$ schedulers.

Figure 26: **Using schedulers $\hat{\eta}^{(t)}$ and $\hat{\alpha}^{(t)}$ with the convex combination formulation is the same as using the original AdEMAMix.** The original AdEMAMix forumulation uses a cosine and linear schedulers for resp. $\eta$ and $\alpha$. Those schedulers are in blue. For the weights given to $\boldsymbol{m}_1$ and $\boldsymbol{m}_2$ to be the same when using the convex combination formulation, one need to use different schedulers $\hat{\eta}^{(t)}$ and $\hat{\alpha}^{(t)}$ (in orange). Using the original AdEMAMix formulation allows to use simpler schedulers.

### C.1.10 MISCELLANEOUS RESULTS

**Impact of $\beta_3$ and $\alpha$ schedulers when starting AdEMAMix from AdamW.** As mentioned in § 3, we do not necessarily need the $\alpha$ and $\beta_3$ schedulers when resuming training from a sufficiently late checkpoint. Fig. 5b and Fig. 5c do not use schedulers, unless we start using AdEMAMix from scratch. In Fig. 27, we vary the warmup periods for $\alpha$ and $\beta_3$ when starting AdEMAMix from an AdamW checkpoint at $t_{switch} = 300k$. We observe how increasing $T_{\beta_3} = T_\alpha$ only slows down the convergence. This serves to illustrate that the schedulers for $\alpha$ and $\beta_3$ are only required to stabilise the optimization during the early training phase.

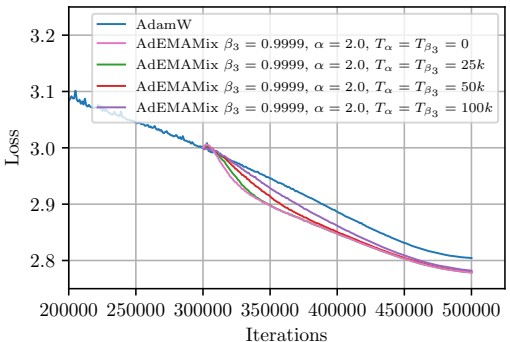

Figure 27: **No need to use schedulers when switching from AdamW to AdEMAMix sufficiently late.** For a 110M parameter model, we switch from AdamW to AdEMAMix after $t_{switch} = 300k$ iterations. We vary $T_{\beta_3} = T_\alpha \in \{0, 25k, 50k, 100k\}$, the other prameters are fixed: $\beta_3 = 0.9999, \alpha = 2$, the learning rate is inherited from AdamW and stays on his decaying course following a cosine decay. AdEMAMix's $\boldsymbol{m}_2$ is initialized to $\boldsymbol{0}$. The larger $T_{\beta_3} = T_\alpha$, the slower the convergence is, the final loss value are very similar. This shows that the schedulers for $\alpha$ and $\beta_3$ are not important when starting AdEMAMix from a sufficiently late checkpoint.

**AdEMAMix from scratch without schedulers.** In this section we provide more justification for the use of schedulers on $\alpha$ and $\beta_3$. We reveal that, without the use of schedulers, the norms of the updates increases significantly, even for small $\alpha$ values. Unstable and large weight updates are known to occur in the early phases of training when learning rate warmup is not used (Gotmare et al., 2019). Using large momentum values too early seem to increase this phenomenon. See Fig. 28.

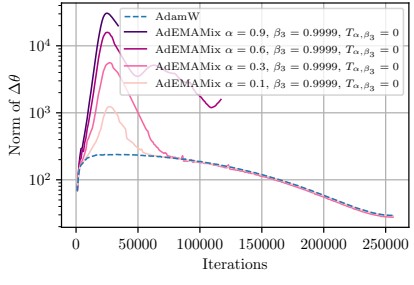

(a) Norm of 1000 consecutive updates.

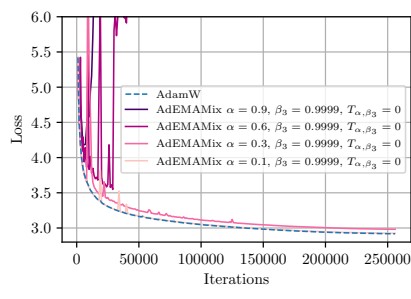

(b) Loss.

Figure 28: **The norm of the updates is exploding without schedulers.** For historical reasons, those experiments were done with a 110M parameter model using a batch size of 32, and without weight decay ($\lambda = 0$). We use $\beta_2 = 0.999$ and $\beta_1 = 0.9$. We use 3000 steps of learning rate warmup for all the experiments. In **(a)**, we monitor the norm of 1000 consecutive updates: $\|\Delta\boldsymbol{\theta}^{(t)}\|_2 = \|\boldsymbol{\theta}^{(t+1000)} - \boldsymbol{\theta}^{(t)}\|_2$. Even when using small values of $\alpha$, the norm of $\Delta\boldsymbol{\theta}^{(t)}$ is increasing significantly at the beginning of training. As a result, in **(b)**, we observe how—without the use of a scheduler on $\alpha$ and $\beta_3$—AdEMAMix models are either diverging or fail to recover and improve upon AdamW.

**Impact of the linear decay duration when using a constant $\eta$-scheduler.** In Fig. 3b we show results using AdEMAMix with a linear warmup $\rightarrow$ constant $\rightarrow$ linear decay learning rate scheduler. We used $100k$ of linear decay. In this section we experiment with $200k$ steps of linear decay, the rest of the parameters are the same: we use a 1.3B parameter model with a max learning rate of $10^{-4}$ and remaining hyperparameters as in Table 4. Results can be seen in Fig. 29.

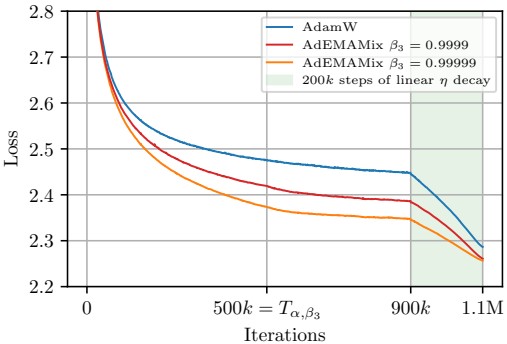

Figure 29: **Testing a different decay duration compared to Fig. 3b.** Increasing the length of the learning rate decay improves the results for each methods. The gap between AdEMAMix using $\beta_3 = 0.99999$ and $\beta_3 = 0.9999$ is shrinking, but the gap between AdEMAMix models and AdamW remains.

**Same figure as Fig. 1, including the AdamW trained on 197B tokens.** In Fig. 1 we represent the AdamW experiment trained on 197B tokens by a blue horizontal line for aesthetic reasons. In Fig. 30 we include this missing curve to the same plot.

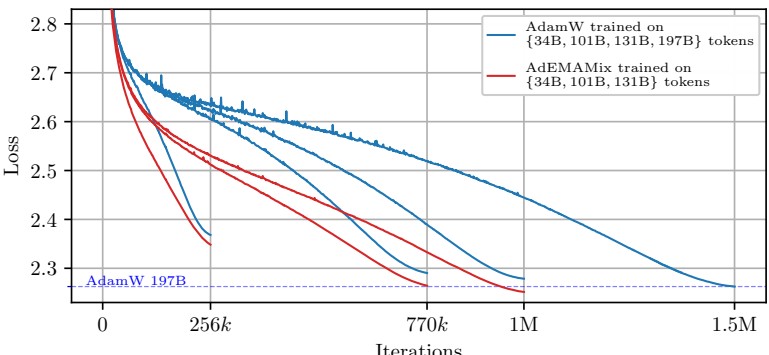

Figure 30: **Same as Fig. 1 adding AdamW 197B.**

## C.2 VιT EXPERIMENTS

**Top-k accuracies.** In Fig. 6, we plot the test and train loss for the final iterates. In Fig. 31, we give a more detailed view by reporting the evolution of the training loss, test loss, and top-1 accuracy. Looking at the first row, the training loss for AdEMAMix is systematically better than the AdamW baseline. The second row shows that in cases where the test loss correlates well with the train loss, AdEMAMix works well. The top-1 accuracy carries the same message.

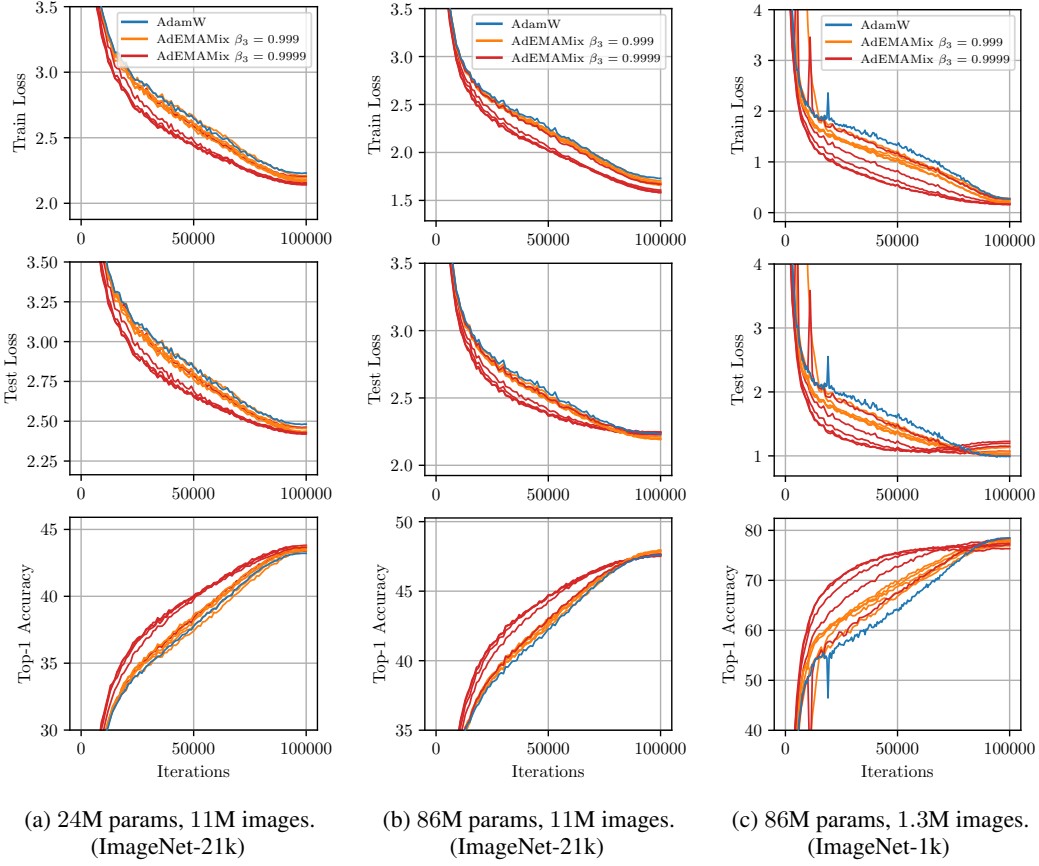

(a) 24M params, 11M images. (ImageNet-21k)

(b) 86M params, 11M images. (ImageNet-21k)

(c) 86M params, 1.3M images. (ImageNet-1k)

Figure 31: **AdEMAMix & capacity/data ratio.** These figures complement Fig. 6 and provide the top-1 accuracy as well as the training and test losses for the three experimental setups used in our ViT experiments (see § 4.3). From left to right, the ratio between data and model capacity worsen. In **(a)**, we train a 24M parameter model on a dataset of 11M images (Imagenet-21k), doing a total of 37 epochs. In this setting the training and test loss curves are very similar, signaling no blatant overfitting. We observe AdEMAMix trivially outperforms the AdamW baseline. As the ratio between data and model capacity worsen, in **(b)** (more capacity) and **(c)** (more capacity and less data), it becomes more and more difficult to find hyperparameters outperforming the baseline. Yet, across all those settings, the training loss for AdEMAMix is always lower than that of the AdamW baseline. When decreasing training loss correlates well with decreasing test loss, AdEMAMix can be expected to work well.

### C.3 COMPARISON WITH OTHER METHODS

#### C.3.1 COMPARISON WITH ADMETA AND DEMA

**Double Exponential Moving Average (DEMA).** Originally introduced by Mulloy (1994), a DEMA originally aimed to emphasize the weight of recent asset price fluctuations, making the DEMA indicator more reactive to changes compared to simple EMAs. Given notations introduced in the main paper, let $\text{EMA}_\beta^{(T-N:T)} \triangleq \text{EMA}(\beta, \boldsymbol{g}^{(T-N)}, \dots, \boldsymbol{g}^{(T)})$, let $N$ be the window size representing how many consecutive values are considered in the average, the formula for DEMA can be written as follows:

$$\text{DEMA}(\beta, \boldsymbol{g}^{(T-2N)}, \dots, \boldsymbol{g}^{(T)}) = 2 \cdot \text{EMA}_\beta^{(T-N:T)} - \text{EMA}(\beta, \text{EMA}_\beta^{(T-2N:T-N)}, \dots, \text{EMA}_\beta^{(T-N:T)}).$$

If a simple EMA tends to give a significant weight to more recent observations, a DEMA emphasizes this behaviour by removing some of the weight given to older observations. This is not what we suggest doing in this work, we want both high sensitivity to recent observations and non-negligible weights given to older observations.

**AdMeta.** Chen et al. (2023b) take inspiration over DEMA and use nested EMAs in their AdMeta-S optimizer:

$$\begin{cases} \boldsymbol{m}_1^{(t)} &= \beta_1 \boldsymbol{m}_1^{(t-1)} + \boldsymbol{g}^{(t)} \\ \boldsymbol{h}^{(t)} &= \kappa \boldsymbol{g}^{(t)} + \mu \boldsymbol{m}_1^{(t)} \\ \boldsymbol{m}_2^{(t)} &= \beta_2 \boldsymbol{m}_2^{(t-1)} + (1 - \beta_2)\boldsymbol{h}^{(t)} \\ \boldsymbol{\theta}^{(t)} &= \boldsymbol{\theta}^{(t-1)} - \eta \boldsymbol{m}_2^{(t)}. \end{cases} \quad \text{(AdMeta-S)}$$

With $\mu$ and $\kappa$ parameterized by $\beta_1 \in [0, 1[$ as such:

$$\mu = 25 - 10(\beta_1 + \frac{1}{\beta_1})$$

$$\kappa = \frac{10}{\beta_1} - 9.$$

In their AdMeta-S experiments, they use $\beta_1 = 0.9$, corresponding to $(\mu, \kappa) = (4.88, 2.11)$. $\beta_2$ takes values ranging from $0.1$ to $0.4$. As a results, unlike AdEMAMix, AdMeta is not leveraging very old gradients. Analysing the AdMeta algorithm, we see that it consists in two nested EMAs. We show the shape of nested EMAs in Fig. 32. In sharp contrast with our approach we observe that (i) it reduces the weights given to recent gradients and (ii) it gives a small weight to old gradients.

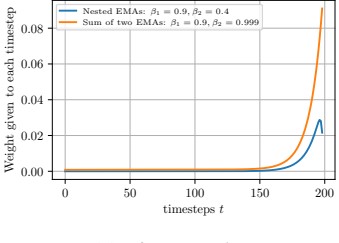
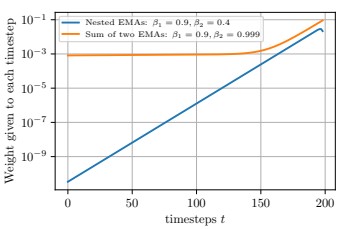

(a) Linear scale.       (b) Logarithmic scale.

Figure 32: **Comparison between nested EMAs and a linear combination of EMAs.** In **(a)** we observe how two nested EMAs actually decrease the influence of recent timesteps. This is the justification for the DEMA method, which aims to increase the sensitivity to recent gradients by subtracting nested EMAs. In **(b)**, using a log-scale for the $y$-axis, we see how, for the values used in AdMeta, the older gradients receive a negligible weight. In our work, we show how very old gradients can be leveraged to get better results by keeping a high sensitivity to recent gradients while giving non-negligible weights to older gradients.

### C.3.2 COMPARISON WITH LION

**The Lion optimizer.** Chen et al. (2023a) derived the following optimizer. We change notations to facilitate the comparison with AdEMAMix:

$$\begin{cases} \boldsymbol{\theta}^{(t)} & = \boldsymbol{\theta}^{(t-1)} - \eta \cdot \left( \text{sign}(\alpha \boldsymbol{m}^{(t-1)} + (1 - \alpha)\boldsymbol{g}^{(t)}) + \lambda \boldsymbol{\theta}^{(t-1)} \right) \\ \boldsymbol{m}^{(t)} & = \beta \boldsymbol{m}^{(t-1)} + (1 - \beta)\boldsymbol{g}^{(t)}. \end{cases} \tag{Lion}$$

The Lion optimizer uses a sign function and updates its EMA after updating the parameters. Moreover, it does not normalize the updates. While Lion and AdEMAMix are quite different from each others, we can draw one similarity. Indeed, the interpolation $\alpha \boldsymbol{m}^{(t-1)} + (1 - \alpha)\boldsymbol{g}^{(t)}$ is similar to combining two EMAs, one of them using $\beta = 0$. We also explore the possibility of setting AdEMAMix's $\beta_1$ to $0$ in § 3, we show in App. C.1.8 and App. C.1.5 (Fig. 16b) that this often works despite a more unstable training. Interestingly, Chen et al. (2023a) find that larger $\beta = 0.99$ values work best. Beside this similarity, the two optimizers behave very differently, the biggest difference being the use of the sign function, which Chen et al. (2023a) claim can help regularize the training. We test the Lion optimizer on language modeling. To tune the hyperparameters, we took values from Chen et al. (2023a) as a starting point, as well as recipes provided by the Optax Jax library. A summary of our hyperparameter tuning is in Table 8:

Table 8: **Hyperparameter tuning for Lion** 110**M parameter models.** When multiple values are given, we bold the parameters we found to give the best results.

| Hyperparameter | Value |
|---|---|
| Learning rate $\eta$ | $0.00005, 0.0001, 0.0002, \mathbf{0.0004}, 0.0006, 0.0008, 0.001$ |
| Number of warmup steps | 3000 |
| Sequence length | 1024 |
| Weight decay $\lambda$ | $0.01, 0.125, 0.166, \mathbf{0.25}, 0.5, 1.0$ |
| Learning rate decay scheduler | cosine-decay |
| Batch size | 64 |
| Gradient clipping | $1.0, \mathbf{0.5}$ |
| Lion $\alpha$ | $0.5, 0.7, \mathbf{0.9}$ |
| Lion $\beta$ | $\mathbf{0.99}, 0.9999$ |

The training curve associated to the best hyperparameters is in Fig. 33. We observe that Lion is not outperforming our carefully tuned AdamW baseline. Moreover, no attempt to increase $\beta$ beyond $0.99$ was successful, as those models mostly diverged. This emphasizes one of our main contributions: the introduction of schedulers enabling the use of large momentum values.

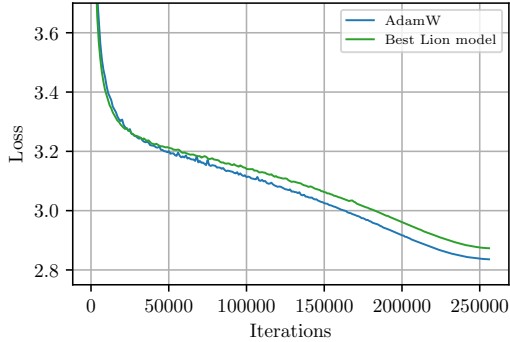

Figure 33: **Comparing AdamW with the best Lion model found.** Hyperparameters used are in Table 8. We train many Lion models and select the one with the lowest final validation loss. Lion is not outperforming our carefully tuned AdamW baseline.

### C.3.3  ADDING A THIRD MOMENTUM TERM (∼AGGMO)

Lucas et al. (2019, AggMo) propose to add an arbitrary number $(K)$ of momentum terms to the gradient descent algorithm:

$$\begin{cases} \boldsymbol{m}_i^{(t)} & = \beta_i \boldsymbol{m}_i^{(t-1)} + \boldsymbol{g}^{(t)} \quad \text{for } 1 \le i \le K \\ \boldsymbol{\theta}^{(t)} & = \boldsymbol{\theta}^{(t-1)} - \frac{\eta}{K} \sum_{1 \le i \le K} \boldsymbol{m}_i^{(t)}. \end{cases} \tag{AggMo}$$

In order to see whether more momentum terms can be beneficial, we modify our AdEMAMix optimizer to include a third momentum $\boldsymbol{m}_3$. We name the resulting optimizer Ad**3**EMAMix:

$$\begin{cases} \boldsymbol{m}_1^{(t)} = \beta_1 \boldsymbol{m}_1^{(t-1)} + (1-\beta_1)\boldsymbol{g}^{(t)}, \qquad \hat{\boldsymbol{m}}_1^{(t)} = \frac{\boldsymbol{m}_1^{(t)}}{1-\beta_1^t} \\ \boldsymbol{m}_2^{(t)} = \beta_3 \boldsymbol{m}_2^{(t-1)} + (1-\beta_3)\boldsymbol{g}^{(t)} \\ \boldsymbol{m}_3^{(t)} = \beta_4 \boldsymbol{m}_3^{(t-1)} + (1-\beta_4)\boldsymbol{g}^{(t)} \\ \boldsymbol{\nu}^{(t)} = \beta_2 \boldsymbol{\nu}^{(t-1)} + (1-\beta_2)\boldsymbol{g}^{(t)^2}, \qquad \hat{\boldsymbol{\nu}}^{(t)} = \frac{\boldsymbol{\nu}^{(t)}}{1-\beta_2^t} \\ \boldsymbol{\theta}^{(t)} = \boldsymbol{\theta}^{(t-1)} - \eta\big(\frac{\hat{\boldsymbol{m}}_1^{(t)}+\alpha(\boldsymbol{m}_2^{(t)}+\boldsymbol{m}_3^{(t)})}{\sqrt{\hat{\boldsymbol{\nu}}^{(t)}}+\epsilon} + \lambda\boldsymbol{\theta}^{(t-1)}\big). \end{cases} \tag{Ad3EMAMix}$$

We Train a 110M parameter model with same hyperparameters as in Table 2, but we use $\alpha = 4$ instead of $\alpha = 8$. We apply the same scheduler to $\beta_4$ as for $\beta_3$. In Fig. 34 we show the resulting validation loss curves for various $(\beta_1, \beta_3, \beta_4)$ triplets. Our experiments show no advantage of adding an extra $\beta_4$-EMA. Rather than carrying the message that we should use more momentum terms, our work shows how—in order to use a large momentum—a term should be added to stay sensitive to local fluctuations of the loss landscape. In that sense, we believe two terms should be enough.

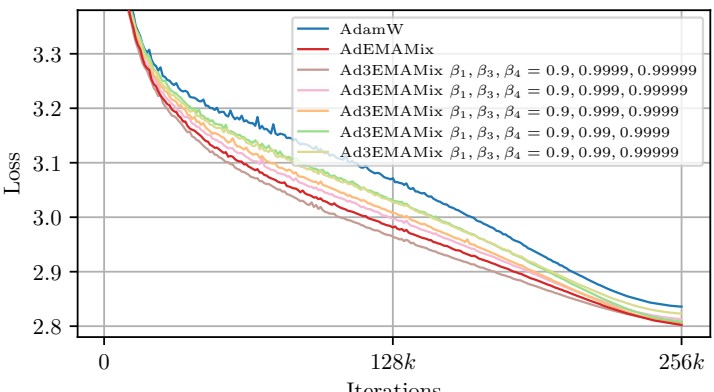

Figure 34: **Adding an extra momentum to AdEMAMix does not provide better results.** We train multiple 110M parameter Ad3EMAMix models with various $(\beta_1, \beta_3, \beta_4)$ triplets (lighter curves). None of those models end up outperforming AdEMAMix, indicating that the additional $\boldsymbol{m}_3$ momentum term is seemingly not bringing any advantage.

