# OpenReview forum: "The AdEMAMix Optimizer: Better, Faster, Older"
_ICLR.cc/2025/Conference — ICLR 2025 Poster_

### Official Review · Reviewer_A3u9 · 2024-11-04

**Soundness:** 3
**Presentation:** 2
**Contribution:** 3
**Rating:** 6
**Confidence:** 4

**Summary:**

This work proposes a new optimization algorithm named AdEMAMix, a variant of Adam(W). They add an aditional moving average of gradients to the algorithm, on top of the existing first and second moment moment averages. In contrast to the traditional  $\beta_1 = 0.9 $ moving average typically used in Adam, the new moving average that updates much slower ( $\beta_3 = 0.9999 $). This additional moving average is then used with the traditional moving average to update the parameters, in a similar fashion to the Adam update. The authors argue this assists in preventing the “forgetting” of data and better uses past information than the traditional Adam update, and show strong empiracle performance on some vision and language settings.

**Strengths:**

The primary strength of this paper apperas to be an algorithm with strong empiracle performance. The authors conduct a thorough set of experiments and ablation in the setting of language modelling and vision using multiple architectures and configuarations. The idea is interesting and combines other ideas from the optimization literature (the Adam update, multiple momentum terms). I am particularly interested in the authors study of how the loss with respect to different batches changes over time depending on the order in which they where trained on. I think this direction is a promising area of research for understanding why some optimizers work better than others. The paper also contains many interesting questions that are of interest to the optimization community, although struggles to answer them (as does the rest of the field).

**Weaknesses:**

Some specific concerns are addressed in the questions section, but I think the fundamental challenge this paper faces is explaining why the algorithm is better. As is typical with deep learning optimizers, it would be too hard for any provable reason why the algorithm may be superior, but I do think some more effort in the paper could be focused on explaining why this is a good idea, especially with the increased memory requirments. As said in the strengths, I do like the amount of questions being asked but it would be nice to see if the answer to any of those questions is leading to a better optimizer. I think it’s possible that the “forgetting slower” direction may lead to an answer, but the majority of the experiments and argumentation for that is relegated to the appendix. While practically showing plots with better performance is good, from a scientific perspective understanding why that performance is better is important.

I’m slightly concernerd by the number of additional hyperparameters being introduced, which appear to be somewhat sensitive given the use of schedules for most of them. This one can make the algorithm more difficult to use in practice, and two runs the risk of having some lucky configurations for a given problem that works great but in general may not be great. I also feel that the additional memory requirment is a non trivial drawback. While it does not increase communication costs, just running in FSDP is not an easy option for many people who do not have easy access to multi-gpu servers.

A more cosmetic note, the figures in this work can lead to confusion at times. In multiple of the figures, there are three subplots where some and perhaps all are unrelated. In some cases, Some subplots are meant to be related, but use different colours for the same thing. This is also likely the reason many of the figure captions are  $\approx $ half a page. I think it would be valuable for the authors to give the plots another pass before any camera ready version.

Overall despite being a potentially effective algorithm, I do feel that this paper is somewhat lacking in the motivation and justification of that. I would be open to raising my score in the discussion period but in it’s current state I’m not sure this work is ready. I would be interested in why the authors think this is a good idea and if there are better ways to show that to be the case. This does not need to be large experiments or rigorous theory. Well designed small experiments or simple derivations can be just as effective give an intuition to explain the performance on minimizing the loss or solving some problem existing optimizers have.

More specific questions can be found below.

**Questions:**

- In figure 2, authors claim the proposed algorithm does not exhibit oscilations when compared to Adam. However, in both 2 (a) and 2 (c) it appears the AdEMAMix does in fact overshoot the minimizer, further than Adam at that. It looks to me that Adam with “default” hypterparamters appears to be the most stable of all the options. In general this is a little concerning as modern loss surfaces will be much more complex than the Rosenbrock function and these large oscilations/overshooting minimizers may be difficult to deal with or even diagnose. A seperate point on this figure but potentially contributing to my confusion is that the colour scheme in the three subplots is not consistent, if the authors could use a shared legend or something along those lines the results may be more interpretable.

- Given this method requires an additional buffer for the second EMA, this authors discuss it potentially being slower and mention that it is also possible to set  $\beta_1 $ to zero to compensate for this. I’m curious if any of the main experiments or the wall time comparison are using this  $\beta_1= 0 $. The hyperparameter configuration for figure 1 uses  $\beta_1 =0.9 $ so I would just  want ensure the wall time comparison is too to ensure figure 1 doesnt change substantially if the x axis is changed from steps to hours.

- What does older gradients being outdated mean? Is there any way to quantify this? Any intuition? All the data is weighted equally as far as minimizing the loss is concerned despite being at different points in the loss surface when you evaluate it. In general this whole idea of older/newer gradients being less/more relevant seems vaugly okay, but it would be nice to see a more scientific defintion or explanation. This doesn’t need to be totally rigorus and have proof but the current explanations seem hand-wavey and leads to far more questions than answers.

- Why not just use a normal non exponential moving average? You can have a uniformly weighted moving average that should not diverge, has this been tried?

- Is there any intuition for why  $\alpha $ is so large (in particlar  $\ge1 $?) Does this lead to a smaller step size being necessary? Can you also set it to zero until some point to fight early instability?

- Why is there no bias correction on  $m_2 $? I don’t really think it matters much much and most analysis ignores it but I’m curious why that decision was made in comparison to the typical Adam bias correction step.

- Regarding figure 4, while it does appear that the proposed algorithm “forgets” slightly less quickly than AdamW, is the claim that this is why it works better? It appears to me that the biggest predictor of “forgetting” is the injection time  $t_B $ rather than the algorithm, although perhaps if this minor difference is compounded across batches this leads to the superior performance of the algorithm.

- In many experiments, notably figure 5, the x axis is limited to not show the optimization performance in earlier parts of training. Given that instability is cited as a challenge for this algorithm, to a point where complex schedules need to be used, I’m curious how unstable the well tuned algorithm is. Are there instability issues in the 0-200k range in figure 5 (b) with AdEMAMix from 0 for example?

- Recent analysis in [1] has shown Adam(W) should work well if the gradient and Hessian become correlated, and [2] provided a mechanism where that shows up in language models. Can the authors reconcile this with their modifications to the algorithm? It seems to me those works would suggest this new variant is not a great idea so I’m wondering if the authors have any thoughts for why that isn’t the case.

- Any idea why there is less good performance on imagenet 1k? It seems AdamW is performing better there although all the plots are fairly close.


[1]
Michael Crawshaw, Mingrui Liu, Francesco Orabona, Wei Zhang, Zhenxun Zhuang

Robustness to Unbounded Smoothness of Generalized SignSGD

https://openreview.net/forum?id=8oj_2Ypp0j

[2]
Frederik Kunstner, Robin Yadav, Alan Milligan, Mark Schmidt, Alberto Bietti

Heavy-Tailed Class Imbalance and Why Adam Outperforms Gradient Descent on Language Models

https://arxiv.org/abs/2402.19449

---

> ### Author Response · Authors · 2024-11-21
>
> We thank the reviewer for the time taken to review our work. On the intuition behind our method. We provide an intuition in our Rosenbrock experiments. In those experiments, the fast EMA (m1) helps the algorithm stay in the valley, and the slow EMA (m2) helps the algorithm crawl along the valley, remembering the valley's direction and accelerating. As pointed out by reviewer GCtm, a recent line of work [1] suggests a “river-valley” structure of the loss landscape of LLMs, which does share some similarities with the Rosenbrock function. We provide another intuition through studying the forgetting of the training data during training (Fig.4). This shows how the larger momentum can help the algorithm to be more data-efficient.
>
> Now answering the reviewer’s questions:
>
> 1. In Fig.2.a, we indeed observe the distance to the solution going down and then up again. This is due to the momentum values causing the iterates to overshoot the solution. When we mention the lack of oscillations for AdEMAMix, we refer to the trajectory not oscillating around the bottom of the valley. The inability of Adam to follow a similar trajectory when using a large $\beta_1$ is what we want to focus on. It shows that—for larger momentum values—the iterate fails to respond to local variations of the loss landscape. This motivates us to also include a momentum term with a small $\beta$ as in AdEMAMix. The oscillations of momentum methods around the solution on deterministic objectives are well documented, see e.g. https://francisbach.com/continuized-acceleration/, https://distill.pub/2017/momentum/, or [2, fig5]. In practice, to the best of our knowledge, oscillations due to momentum have never been an issue in deep learning training (i.e. the phenomenon of overshooting the solution seems to be only observed in toy settings, see Fig.20 and Fig.21 to see the lack of oscillations even with very large $\beta_1$ in a real world setting using Adam). This is why we focus on the left part of the curves, corresponding to the first time required to arrive in the vicinity of the solution.
> 2. The times in Fig.5.a use $\beta_1=0.9$.
> 3. Given a past timestep $t$, a sample $x$, the gradient $\nabla_\theta \ell(x, \theta_{t})$ is outdated at time $t+K$ if its inner product with $\nabla_\theta \ell(x, \theta_{t+K})$ is non-positive. Such a gradient would be outdated as it pushes the iterate in a direction that is no longer relevant to decrease the loss on $x$. A simple illustration could be overshooting the minimum of a quadratic function. Gradients are local approximations, we mention outdated gradients as it is surprising for us that one can take advantage of very old gradients. Our intuition was that those would no longer be relevant given the local loss landscape. We believe that the fact this raises many questions is a good thing.
> 4. In a limited setting, we also tried with uniform averages, we found those to also improve over the baseline but much less. We therefore decided to focus on EMAs.
> 5. We do use a linear scheduler on alpha. Starting from $0$, alpha grows up to its final value over the same number of iterations as $\beta_3$. Setting it to $0$ during the early iterations can work, this corresponds to our experiments on switching from AdamW to AdEMAMix (Fig.5.b and Fig.5.c). Depending on when the switch occurs, the final loss—while better than the baseline—might not be as good as training AdEMAMix from scratch. In Fig.5.b and Fig.5.c, we do not use any schedulers on beta3 nor alpha.
> 5. The bias correction is there to compensate for the initialization of the buffers to $0$. In our case, it is desirable to not do the correction as it implies the values of m2 are initially very small, and increase little by little. Intuitively, this means that, for the early steps, AdEMAMix behaves similarly to AdamW. Moreover, when switching from AdamW to AdEMAMix, the effective learning rate increases, which we believe explains the small increase in loss that can be observed right after the switch in Fig.5.b and Fig.5.c. Not doing any bias correction might help to smooth the transition.
> 7. We were curious on what was explaining the correlation between forgetting and iterations and ran additional experiments in App.C.1.3 (see Fig.11). In that experiment we use a WSD learning rate scheduler instead of a cosine decay. This allowed us to show that the forgetting is tied not only to the optimizer, but also to the learning rate. Setting this dependency aside, it is clear from Fig.4 that AdEMAMix forgets training batches slower than AdamW.

---

> > ### Author Response · Authors · 2024-11-21
> >
> > 8. In Fig.5, the curves partially shown for AdamW and “AdEMAMix from scratch” are similar to the ones in Fig.1. Curves for Fig.5.b reach similar losses as the curves in Fig.1.a trained for 500k iterations. Similarly, the two curves from Fig.5.c reach similar losses as the curves in Fig.1.c trained for 770k iterations. For historical reasons, Fig.5 relies on a larger clipping, which does not affect the performance but smoothes the curves. We thank the reviewer for raising this point and will update the description of our experimental protocol to clarify this point.
> > 9. We thank the reviewer for sharing those interesting works. We would appreciate it if the reviewer could elaborate on which part of those works suggests that our method should not work well?
> > 10. On imageNet 1k, Fig.31 might paint a clearer picture of what is happening. Looking at the test and training losses on the last column, we see that while the training and test losses decrease faster for AdEMAMix, the test loss shows a clear overfitting pattern. We believe AdEMAMix to be best suited in cases where enough data is available w.r.t. the capacity of the model.
> >
> > We hope we have addressed the concerns of the reviewer. We stay at their disposal for any further questions and hope the above response would bring the reviewer to raise their score.
> >
> > [1] Wen et al., 2024: “Understanding Warmup-Stable-Decay Learning Rates: A River Valley Loss Landscape Perspective”
> > [2] Flammarion et al., 2015: "From averaging to acceleration, there is only a step-size."

---

> ### Comment · Reviewer_A3u9 · 2024-11-27
>
> I thank authors for their responses. In regards to their question about why the work I cited may not agree with their approach, I preface  by saying this is my  inutition and not rigorous. In [1], it is shown that Adam will benefit when the norm of the gradients and the trace of the Hessians become correlated. Then, [2] shows a situation where this occurs naturally where Adam  is superior to SGD. My intuition is that, since now we have two EMAs (with the slower changing one being more dominant due to $\alpha\ge1$) for the gradients but only one for the squared  gradient, this correlation will be weakened. In future work, it may be a good experiment to run in order to figue out if this algorithm is effective for similar  reasons to Adam. If the authors  have any thoughts on this I would be interested.
>
> Overall I do still feel that there could more investigation of why this algorithm works better even with the response probvided by the authors. I will  raise my score to a weak accept based on the authors responses and hopw that that question will be addressed in future work. I would again like to point out that some of the figures a bit confusing as is stated  in my original review,  so if this work ends up getting accepted I reccommend the authors take  a  second  pass on  them.

---

### Official Review · Reviewer_98Nr · 2024-11-04

**Soundness:** 3
**Presentation:** 4
**Contribution:** 3
**Rating:** 6
**Confidence:** 4

**Summary:**

This paper introduces AdEMAMix, an optimizer modifying Adam by incorporating a mixture of two Exponential Moving Averages (EMAs) to better leverage past gradients. Traditional Adam or AdamW optimizers rely on a single EMA to smooth gradients, which prioritizes recent gradients over older ones. However, the authors argue that this approach is suboptimal because it cannot simultaneously emphasize both recent and very old gradients. They conduct experiments across language (LLMs and Mamba) and vision (ViTs), showing that AdEMAMix outperforms and tends to forget training data slower than Adam.

**Strengths:**

- The paper is written and organized well. The optimizer is a simple modification to Adam and is conceptually easy to understand, with the algorithmic differences presented clearly.
- The authors extensively benchmark their proposed optimizer against Adam across architectures, dataset domains and hyperparameters.
- The spirit of AdEMAMix (having a ‘fast’ and ‘slow’ tracking of gradients) could be useful for mitigating forgetting in continual learning regimes, and is also shown to be relevant for current pretraining regimes where we often train much longer past eg. Chinchilla-optimal tokens.
- The authors have experiments which address one’s immediate concerns of the proposed optimizer, like hyperparameter tuning and memory overhead.

**Weaknesses:**

- The proposed optimizer is not that novel, with several existing approaches in maintaining a longer horizon at the cost of memory which have been mentioned by authors; in particular, AggMo is essentially the same as AdEMAMix but applied to gradient descent, and the only difference in the setting of this work is applying the same principle to Adam.
- Although the authors have performed experiments on the hyperparameter stability of AdEMAMix, they do introduce two new parameters beta_3 and alpha which require the use of schedulers for larger values to avoid divergence in early iterations, which may require more extensive tuning depending on the setting (eg. fast domain shifts).
- There is an additional memory overhead in the order of the model size to incorporate the second EMA, which the authors propose can be mitigated by setting beta_1 to be 0; however, these runs exhibit instabilities and sometimes large spikes in training especially at larger batch sizes (Figure 22). These spikes are generally undesirable even though the loss seems to ‘recover’, it is unclear to me that such spikes wouldn’t have more detrimental effects at larger scales.

**Questions:**

1. Doesn’t the introduction of the additional beta_3 term potentially lead to an exploding step size if the incoming gradients are extremely small for many steps (the additional term in the numerator causes it to decay slower than the denominator in the update)? This isn’t unforeseeable in language model training, where the model could encounter multiple documents of rare tokens consecutively.
2. Do the authors have any experiments adding an EMA to other optimizers like Adafactor or Signum? Does the additional EMA always provide a gain, even with factored gradients?
3. Related to the previous question to address the memory overhead, does storing the slow-moving EMA in lower precision affect AdEMAMix performance?
4. Why was the optimal Adam learning rate being too high for AdEMAMix only manifesting at higher scales? Does this perhaps suggest AdEMAMix being more difficult to tune at larger scales? I’m wondering if there was there further investigation into this eg. Was there a certain part of the network that was destabilizing?
5. Did the authors use certain stabilization strategies, eg. QK-LayerNorm in your experiments? I’m wondering if this could be used for the author’s benefit, to stabilize training and offer greater ease in hyperparameter tuning.
6. Why was alpha set to 2 and 1 for the switching Adam -> AdEMAMix experiments whereas the optimal values found in hyperparameter tuning were much higher?

---

> ### Author Response · Authors · 2024-11-21
>
> We thank the reviewer for the time taken to review our work. First, responding to the weaknesses identified by the reviewer:
>
> 1. On the novelty of our approach compared to AggMo. While the AggMo method introduced by Lucas et al. also adds additional EMAs with larger beta values, we strongly disagree that it implies our work lacks novelty. As discussed in the related work, the claims from Lucas et al. are that additional momentum terms speed up convergence and hyperparameter stability. In our work we show our optimizer is not only converging faster, but reaches better solutions (lower loss). Moreover, Lucas et al., relies on small-scale settings where SGD works well, using MNIST, CIFAR-10 and CIFAR-100, and training LSTM LMs on the Penn Treebank dataset. In contrast, we focus on significantly larger real-world settings. Those larger settings heavily favor Adam over SGD. Getting our optimizer to work in those settings was not as easy as adding an additional momentum buffer, it required tackling training instabilities through deriving schedulers, without which the method would not work. Finally, AggMo is suggesting using many (e.g. 4) momentum terms. We show in App.C.3.3 that using more than two momentum terms in AdEMAMix is not providing any benefit.
> 2. On using AdEMAMix in fast domain-shift settings. This is a valid concern for that setting. We did not study non-stationary training distributions and leave that question for future work. Concerning schedulers, we observed that our design-choice of always setting $T_{\alpha,\beta_3}=T$ yielded stable training runs without additional validation, even when switching across model size, to state space models and vision models.
> 3. We will change the text to say that using beta_1=0 needs to be confirmed at larger scale. We want to emphasize that, in many cases, the memory overhead is not a critical issue. Especially, in the case of distributed training, the memory overhead can be mitigated by sharding the optimizer state across devices.

---

> > ### Author Response · Authors · 2024-11-21
> >
> > Now answering the reviewer’s questions:
> >
> > 1. While this is possible, we did not observe this in practice. Given the large values of $\beta_2$ and $\beta_3$, the contribution of each individual gradient is very small. For an exploding step size to occur, small gradients would need to be observed over many thousand consecutive steps, which is unlikely if the data is shuffled. Moreover, the epsilon on the denominator can be used to prevent the updates from growing too much. Another related problem is the sensitivity to large gradient norms. Given the large $\beta_3$, outlier gradients can have a long lasting and detrimental effect. In our experiments, we found that using gradient-clipping can be important to mitigate this issue. We thank the reviewer for raising this point and will add a paragraph detailing these two cases in our next revision.
> > 2. We did not try adding an additional EMA to other optimizers like Adafactor or Signum. Our focus was on showing convincingly that our approach works well in standard settings.
> > 3. This is an interesting suggestion. Our work aims to introduce a novel method and show it performs well in standard settings. While we provide some direction to reduce the memory footprint, we do not believe the memory overhead is entirely challenging the usability of the method in many cases. As such, we leave further memory optimizations as future work.
> > 4. For our 1B parameter experiments, when using a learning rate of 5e-4 for AdEMAMix, we observed instabilities in the form of gradient norm spikes. We did not deep dive into our model to check if a certain subpart of the model was causing this. We believe those to be caused by the slight effective learning rate increase resulting from adding alpha*m2 to m1. Our results at a larger 3B scale in App.C.1.2 uses the same learning rate for both AdamW and AdEMAMix.
> > 5. This is a great suggestion. Given our goal to showcase our optimizer in standard settings, we used vanilla architectures in all of our experiments, which do not include QK-LayerNorms. In practice, we did not find it difficult to tune the hyperparameters for AdEMAMix. For instance, results on ViT models in Fig.6 (or Fig.31), show AdEMAMix models trained with the same hyperparameters as the best baseline, using different beta3 and alpha. Given enough data, it is easy to find a combination outperforming Adam.
> > 6. When switching from Adam to AdEMAMix, the effective learning rate suddenly increases, which we believe explains the bump that can be seen right after the switch. Larger alphas can imply larger bumps, which—in our experiments—take longer to recover from. Given more iterations, the model recovers. The values used were small enough to allow the model to recover well given the remaining number of iterations. Interestingly, App.C.1.4 details the reverse experiment, switching from AdEMAMix to AdamW. We observe a drop in loss immediately after the switch, likely explained by a drop in effective learning rate.
> >
> > We hope we have addressed the reviewer’s concerns. We would appreciate it if the reviewer could consider raising their score and stay at their disposal in case any further questions need to be answered.

---

> > > ### Comment · Reviewer_98Nr · 2024-12-03
> > >
> > > I thank the authors for their response and for answering my questions. I have read the other reviews and responses and I do recommend acceptance for the paper. I agree that while the notion of adding more momentum terms is not new, there is value in demonstrating an algorithm which is performant in more practical settings. I will maintain my score as weak accept due to similar comments brought up by the other reviewers; for instance, if the proposed solution for reducing memory overhead by setting beta1 = 0 is sufficient, then why do the authors propose generally having the two momentum signals? Is there an explanation for why setting beta1=0 works? There are also other barriers for its use more generally, eg. in fast domain-shift settings where the question of setting a performant scheduler will likely be more difficult.

---

> > > > ### Author Response · Authors · 2024-12-03
> > > >
> > > > We thank the reviewer for their reply.
> > > >
> > > > 1. We would like to bring some final clarifications on the following point:
> > > >
> > > > > if the proposed solution for reducing memory overhead by setting beta1 = 0 is sufficient, then why do the authors propose generally having the two momentum signals? Is there an explanation for why setting beta1=0 works?
> > > >
> > > > We provide an explanation in our introduction, line 85 to 90, quoted here for convenience:
> > > >
> > > > > We observe that a single EMA cannot both give a significant weight to recent gradients, and give a non-negligible weight to older gradients (see Fig. 3a). However, a linear combination between a “fast-changing” ***(e.g. β1 = 0.9 or β1 = 0)*** and a “slow-changing” (e.g. β = 0.9999) EMA allows the iterate to beneficiate from (i) the great speedup provided by the larger (slow-changing) momentum, while (ii) still being reactive to small changes in the loss landscape (fast-changing).
> > > >
> > > > The core of the method is not to require two EMAs. It is to combine a term gathering information from old gradients ($m_2$), with a term that remains sensitive to local loss variations ($m_1$). When $\beta_1 = 0$, $m_1$ is simply the gradient $g$, so $m_1+\alpha m_2$ becomes $g+\alpha m_2$. Following only the direction of $m_2$ (removing entirely $m_1$), would not work, as shown in our Rosenbrock experiments (Fig.2) and language modeling experiments increasing $\beta_1$ in Adam (Fig.20 and Fig.21).
> > > >
> > > > Now, is there an advantage to using  $\beta_1 = 0.9$ instead of $\beta_1 = 0$? The answer is yes. In App.C.1.8, we show how $\beta_1 = 0$ can be more unstable. For this reason, we kept the two EMA formulation in our method, as there might be cases where $\beta_1 > 0$ is preferred. In general, we recommend using $\beta_1 > 0$ unless memory is an issue.
> > > >
> > > > 2. Concerning your second point:
> > > >
> > > > > There are also other barriers for its use more generally, eg. in fast domain-shift settings where the question of setting a performant scheduler will likely be more difficult.
> > > >
> > > > Our work focuses on a very standard and widely adopted optimization setting, which consists of training models on static datasets. This setting is the one currently used throughout the industry to train models such as large LLMs, state of the art vision models and more. We agree studying our method in domain-shift settings is an interesting direction for future work, yet this is outside of the scope of our work.

---

### Official Review · Reviewer_GCtm · 2024-11-05

**Soundness:** 3
**Presentation:** 3
**Contribution:** 3
**Rating:** 5
**Confidence:** 4

**Summary:**

This paper proposes to use an additional momentum buffer for the Adam optimizer. The motivation behind is that a single momentum buffer may not be able to utilize both past and current gradient information efficiently. To this end, one buffer with large momentum parameter $\beta$ is for slow-changing directions, and the other one with small $\beta$ is for fast adaptation to current gradient. Then, the two momentum buffers are mixed according to some fixed ratio ($\alpha$), and the rest of updates (i.e., pre-conditioning and weight-decay) follow those of AdamW. However, a large $\beta$ (for slow-changing momentum buffer) can cause training instabilities in the initial stage. To this end, the paper proposes to gradually increase this parameter until the target value is reached.  Then, experiments are performed on various language modelling tasks to show that the proposed algorithm is faster than AdamW given a fixed computation budget.

**Strengths:**

- The paper is well written and easy to follow. The experiments on the Rosenbrock function are convincing. There are some similar (loss landscape) models proposed recently to analyze learning rate schemes [1]. It maybe interesting to draw some theory/experimental connections in the case of momentum.

- The experiments seem to be comprehensive covering different settings and tasks. The improvement over baseline is shown.

[1] Understanding Warmup-Stable-Decay Learning Rates: A River Valley Loss Landscape Perspective.

**Weaknesses:**

- There are some additional hyperparameters introduced. Noticeably, it seems that $\alpha$ (that controls the mixing ratio)  is important for the algorithm. It would be important to study the sensitivity of the algorithm to this hyperparameter, given that it essentially controls the contribution of each momentum buffer to the current update.

- It would be better if some convergence guarantees of the algorithm can be provided even in the convex setting. For example, what is the relationship between the two momentum parameters that would guarantee convergence?

**Questions:**

Overall, I think the idea introduced in this paper is interesting. The paper can be improved if the above  weaknesses can be addressed.

---

> ### Author Response · Authors · 2024-11-21
>
> We thank the reviewer for the time taken to review our work.
>
> 1. App.C.1.5 studies extensively the sensitivity to hyperparameters. The sensitivity to the mixing ratio alpha is shown in Fig.13.a. It is shown that the range of alpha values outperforming the Adam baseline is very wide.
> 2. While a theoretical proof of convergence of our method would be welcome, deriving such proof poses significant challenges. We can take as an example Adam, which convergence has been challenged by Reddi et al. [1], which still failed to explain convergence for hyperparameter values used in practice as discussed in [2]. Despite the enduring gap between theory and practice, Adam remained the workhorse of deep learning optimization. Understanding it from a theoretical standpoint is a line of research on its own. As such, we believe providing convergence guarantees—albeit desirable—lies outside the scope of our work. This being said, convergence bounds in convex settings for a simpler method (AggMo) combining GD with a linear combination of EMAs have been shown in [3].
>
> We hope we have addressed the reviewer’s concerns. Sensitivity to hyperparameters is studied in App.C.1.5, and providing theoretical backing seems tedious given a theoretical understanding of even Adam is still an active field of research. We hope the reviewer will consider raising their score or provide further details justifying their rejection of our work.
>
> [1] Reddi et al., 2019: On the Convergence of Adam and Beyond
> [2] Zhang et al., 2022: Adam can converge without any modification on update rules.
> [3] Lucas et al., 2019: Aggregated momentum: Stability through passive damping

---

### Official Review · Reviewer_H5Kk · 2024-11-05

**Soundness:** 4
**Presentation:** 4
**Contribution:** 4
**Rating:** 10
**Confidence:** 4

**Summary:**

This paper proposes AdEMAMix, a new optimizer which outperforms Adam on language model training and ViT training. Their empirical results show large benefits (~50% reduction) over Adam in the regime of noisy gradients i.e. small batch size or longer runs. The main idea behind the optimizer is to maintain two momentum terms and combine them to get the final movement direction. The coefficient of momentum and their combination are also dynamically adapted.

**Strengths:**

These are strong results on a very important problem. They also provide many optimizer ablations in the Appendix showing the robustness of their proposed optimizer.

**Weaknesses:**

Since many of the experiments are with small batch size it would have been interesting to explore the effect of weight averaging. For example, is it the case that weight averaging helps AdamW and AdEMAMix equally? Or not?

**Questions:**

The authors state “While no answer to those questions is given in this work, we
provide a toy justification which indicates that large momentums can have a positive impact in
noise-free non-convex settings (see Fig. 2)—indicating the improvement of our approach is at least partially explainable without considering variance-reduction effects.” Is there empirical support for this in the LLM experiments? looking at Figure 17 the benefit seems to drop with increasing batch size. Note that the maximum batch size used here (512k) is smaller than that used for LLMs like Llama (4m), though I agree that the model size is also small here. Could the authors provide an experiment with 2m batch size to see the trend?

---

> ### Author Response · Authors · 2024-11-21
>
> We thank the reviewer for the positive appreciation of our work.
>
> In Fig.17 we keep the total number of training tokens constant, and vary the number of steps (in {$32k, 64k, 128k, 256k$}). To keep the number of training tokens constant, this forces us to increase the batch size as we decrease the number of steps. Looking at Fig.17.a and Fig.17.b, we observe that both methods suffer when we trade a large number of iterations for a larger batch size and fewer steps. AdEMAMix—while still outperforming Adam—is more affected. To understand this phenomenon, we notice that when we do fewer steps, we have fewer gradients to accumulate in $m_2$. Given our very large $\beta_3$ values (e.g. $0.9999$), this can become a problem. We show in Fig.18 that the problem is mitigated by reducing the $\beta_3$ value to e.g. $0.999$. Interestingly, comparing AdamW in Fig.17.a and AdEMAMix with $\beta_3=0.999$ in Fig.18.b, we can see that AdEMAMix is now less affected than AdamW when we trade a large number of iterations for a larger batch size and fewer steps (the increase in final loss is smaller for AdEMAMix when increasing the batch size). In general, from our experiments on both images and text, we did not notice any disadvantage of ADEMAMix over AdamW when increasing the batch size. Our largest experiments, in App.C.1.2, train 3B parameter models using a batch size of $1024^2$ tokens. We still observe improvements.
>
> We thank again the reviewer for appreciating our work and stay at their disposal to answer any further questions.

---

> > ### Comment · Reviewer_H5Kk · 2024-11-30
> >
> > Thank you for the response, I maintain my positive assement of the work.

---

### Official Review · Reviewer_zVt6 · 2024-11-08

**Soundness:** 3
**Presentation:** 4
**Contribution:** 3
**Rating:** 6
**Confidence:** 4

**Summary:**

The paper propose a new optimizer called AdEMAMix, which add an additional EMA sequence to Adam. The intuition is to to keep a high sensitivity to recent gradients (using m1), while also incorporating information from older gradients (using m2).

**Strengths:**

The prerformance of AdaEMAMix is quite impressive. The writing is excellent. The ablation studies are thorough.

**Weaknesses:**

The motivation is rather vague. See below.

**Questions:**

1. The motivation on the Rosenbrock function is rather weak.  Why do you consider the Rosenbrock function? How does it relate to LLM training?  Why do you think the implication on the Rosenbrock function can be transferred to LLM?  To me, there is a huge gap between these two.

2. Need more evidence that "beta1 = 0.9 is indeed suboptimal for LLMs". While I fully understand Figure 3, i still did not fully convinced that "LLM training really needs to use more historical gradients, in one way or another". More LLM-related evidence is needed to convince the that it is indeed a major bottleneck. Despite the impressive performance, the overall motivation still seems quite vague to me. I suggest the authors put more effort into designing more experiments for better motivation.

3. It is difficult to understand the highlighted sentence in the introduction: "While changing the direction of the slow momentum is difficult, any adjustment orthogonal to that direction is easy—which favors fast progress in sinuous canyon-like landscapes." I don't understand what you mean by "change the direction of the slow momentum" or "any adjustment orthogonal to that direction". Please explain more.  Also, the authors mentioned "canyon-like landscapes (in Rosenbork function)." but never connected it to LLM in the script. This also makes the motivation rather weak.

4. I don't quite understand how to read Figure 2. For instance:

   -- in Figure 2 (a), is AdaEMAMix considered better than Adam? It is not clear whether we can draw such conclusion based on the figure. In my opinion, at least 10 x more iterations are needed to draw valid conclusions.

   -- in Figure 2 (c), did AdaEMAMix converge to the optimal solution?

5. The proposed method boosts performance by using extra memory (to store an additional copy of momentum). Though many readers might regard it as a drawback,  I personally think such a trade-off is acceptable as long as the performance gain is significant.  Further, I think AdaEMAMix can be combined with some orthogonal methods to reduce memory. For instance, AdaEMAMix can be combined with the recent method Adam-mini [1] to reduce the memory for V. I suggest the authors try it out.



6. Some missing related works: [2] proves that vanilla Adam can converge under a wide range of beta1  = 0, 0.5, 0.9, 0.99, etc., as opposed to the divergence result in Reddi et al. 2018. This result lays down a preliminary foundation of this work.  Without the theoretical guarantee, it would be dangerous to play with beta1 of Adam.

[1] Zhang, Y., Chen, C., Li, Z., Ding, T., Wu, C., Ye, Y., ... & Sun, R. (2024). Adam-mini: Use fewer learning rates to gain more. *arXiv preprint arXiv:2406.16793*.

[2] Zhang, Y., Chen, C., Shi, N., Sun, R., & Luo, Z. Q. (2022). Adam can converge without any modification on update rules. *Advances in neural information processing systems*, *35*, 28386-28399.

---

> ### Author Response · Authors · 2024-11-21
>
> We thank the reviewer for taking the time to review our work.
>
> 1. We agree that the gap between Rosenbrock and LLM training is important. We do not claim that this toy setting captures the complexity of LLM training dynamics. Instead, it serves as an illustration, showing that—quite remarkably—our method can be motivated even in a noise-free toy setting. Moreover, while understanding the training dynamics of LLMs is still an active line of research, a recent line of work [1] (also pointed out by reviewer GCtm) suggests a “river-valley” structure of the loss landscape of LLMs, which does share some similarities with the Rosenbrock function. We will cite this work in our next revision.
> 2. We are not exactly sure where our "effort into designing more experiments" should concentrate. Could the reviewer specify the type of experiments that would be needed? We provide two key intuitions that help understand the reason behind the superiority of our approach. First, while limited, our Rosenbrock example shows that using an EMA with a large beta can be beneficial yet needs adjustment of the optimizer (it doesn’t work to simply increase beta1 in Adam). Saying that “$\beta_1=0.9$ is suboptimal for LLMs” is ambiguous, and this is not what we aim to convey. It seems $\beta_1=0.9$ is optimal for Adam (see Fig.20). Using Adam with $\beta_1=0.9999$ does not work, which makes sense as $m_1$ is no longer responsive to local variations of loss landscape, therefore Adam with $\beta_1=0.9999$ fails to optimize the underlying function (Fig.3.a). Our core message is that this can be solved by combining it with a term that stays responsive to the local loss landscape (e.g. an EMA with a small beta). Then, it is possible to gain from using much older gradients. In the Appendix, section C.1.7 is entirely devoted to limitations of using a single EMA in Adam. In that section, we show increasing $\beta_1$ in Adam does not work, even if we bypass the initial training instabilities and start from a pretrained AdamW checkpoint (Fig.21), even when adding schedulers on $\beta_1$ as for AdEMAMix (Fig.20.b). Even when increasing $\beta_2$ from $0.999$ to $0.9999$ to stabilize training (again Fig.20.b). The second intuition we provide relates to our analysis of forgetting (Fig.4). We show that larger beta values as in AdEMAMix can be more data-efficient, improving the final loss on training samples when compared to Adam. While we concede that our understanding of the phenomenon and motivations are not exhaustive, we nonetheless prove empirically that more historical gradients can be used efficiently, while providing several possible justifications.
> 3. The larger the beta, the smaller the contribution of each gradient, and therefore many gradients are needed to change the direction of $m_2$. In contrast, changing $m_1$ only requires a few iterations. Therefore, updating the weights in a direction going against $m_2$ requires pushing against $m_2$ for many iterations. In contrast, updating the weights in a direction orthogonal to $m_2$ is easy. In essence, what we are trying to convey is: the fast EMA ($m_1$) helps the algorithm stay in the valley, and the slow EMA ($m_2$) helps the algorithm crawl along the valley, remembering the valley's direction and accelerating. We will clarify in our next revision.
> 4. Fig.2 aims to show that AdEMAMix can use larger momentum values to reach good solutions faster, with less oscillations. In Fig.2.a, we observe the distance to the solution going down and then up again. This is due to the momentum causing the iterates to overshoot the solution. The oscillations of momentum methods on deterministic objectives are well documented, see e.g. https://francisbach.com/continuized-acceleration/, https://distill.pub/2017/momentum/, or [2, Fig.5]. In practice, to the best of our knowledge, oscillations due to momentum have never been an issue in deep learning training (i.e. the phenomenon of overshooting the solution seems to be only observed in toy settings, see Fig.20 and Fig.21 to see the lack of oscillations even with very large $\beta_1$ in a real world setting). This is why we focus on the left part of the curves, corresponding to the first time required to arrive in the vicinity of the solution. Ultimately, all the methods tested converge to the solution given enough iterations, we only claim that AdEMAMix can find good solutions relatively fast.

---

> > ### Author Response · Authors · 2024-11-21
> >
> > 5. Combining AdEMAMix with Adam-mini is indeed an interesting research direction. In our work, we propose a simpler solution which consists in setting beta1=0. This means that m1 is replaced by the gradient, and the memory cost of AdEMAMix is then equal to Adam. We show in App. C.1.8 that this strategy works in most cases. Investigating more elaborated memory saving strategies is an interesting future work direction. We will cite Adam-mini in our next revision.
> > 6. We thank the reviewer for sharing this work, we will cite it in our next revision.
> >
> > We hope we have addressed the reviewer’s concerns and conveyed the intuition behind our method. We hope the reviewer will consider increasing their score and we stay at their disposal to answer any further questions.
> >
> > [1] Wen et al., 2024: Understanding Warmup-Stable-Decay Learning Rates: A River Valley Loss Landscape Perspective
> > [2] Flammarion et al., 2015: "From averaging to acceleration, there is only a step-size." Conference on learning theory.

---

> ### Comment · Reviewer_zVt6 · 2024-12-01
> **Thanks for the response!**
>
> Thanks for the rebuttal and I sincerely apologize for the late reply. I have carefully read the rebuttal. I still think this is a  **professionally-written paper with strong empirical evidence, yet the motivation is still a bit weak**.  I vote for acceptance and I will keep my score.
>
> Here are some follow-up comments. I think they would be helpful to improve the paper quality.
>
> 1. **Regarding my Q4:** I am clearly aware of "oscillations of momentum methods". This is not my comment. My comment is "it is not clear whether we can draw such a conclusion based on the figure. In my opinion, at least 10 x more iterations are needed to draw valid conclusions.". Anyhow, the authors did not provide new experiments.
>
> 2. **Regarding paper presentation**. Some notations are unclear and are not consistent. For instance, in line 249 "This allows for the use of much larger beta values e.g. 0.9999" what is beta here?  Is it beta1 or beta3?
> 3. **Regarding motivation.** I still think it is too weak to motivate using Rosenbrock function. Here are my suggestions: please try linear model + 2-class cross-entropy loss classification, which also have "river-valley" landscape. Show that AdEMAmix has an advantage on this task (perhaps a bit more theory would be cool), then generalize to 1-hidden-layer-NN + cross-entropy or 1-layer-Transformer + cross-entropy. These experiments and discussions will provide much stronger insight and motivation. At least much better than Rosenbrock.
>
> 3. **Regarding your rebuttal.** In rebuttal, the authors mentioned that "we propose a simpler solution which consists in setting beta1=0. ". This makes me quite confused.  When beta1= 0, how is AdEMAmix different from AdamW with beta1 = 0.9999?  They seem to be the same up to a constant alpha, right? If so, then does it mean that AdamW (beta1 = 0.9999) works better than AdamW (beta1 = 0.9)? However, the authors also claim that AdamW (beta1 = 0.9999) does not work well.  So I am confused what is going on here.
>
>     Further, if AdEMAmix (beta1 = 0) works as well as beta1 >0, then why bother introducing the idea of "balancing fast and slow changing signals"?  It seems that we do not need to balance anything at all. This further makes me confused about the motivation of the paper.
>
> 4. **Missing important discussions.** It is good to have the discussion on line 407 "Why not simply increase AdamW's beta1?" But this discussion is rather numerically guided. I suggest adding a rigorous math clarification that "increasing beta1" is NOT equivalent to "linear combination over an additional EMA copy". (I assume they are indeed different, right? I didn't have time to carefully check)

---

> > ### Author Response · Authors · 2024-12-03
> >
> > We thank the reviewer for their reply. We first would like to focus on addressing a key misunderstanding:
> >
> > >  In rebuttal, the authors mentioned that "we propose a simpler solution which consists in setting beta1=0. ". This makes me quite confused. When beta1= 0, how is AdEMAMix different from AdamW with beta1 = 0.9999? They seem to be the same up to a constant alpha, right?
> >
> > This is not the case. When $\beta_1=0$, $m_1=g$, with $g$ being the gradient. Therefore, $m_1+\alpha m_2$ in AdEMAMix becomes $g + \alpha m_2$. This is quite different from Adam, and still fits our narrative of balancing two signals, one sensitive to local variations of the loss landscape ($g$), and one incorporating information from older gradients ($m_2$).
> >
> > > Further, if AdEMAmix (beta1 = 0) works as well as beta1 >0, then why bother introducing the idea of "balancing fast and slow changing signals"? It seems that we do not need to balance anything at all. This further makes me confused about the motivation of the paper.
> >
> > Now that we have established that AdEMAMix with $\beta_1=0$ is different from Adam, we can ask if there is an advantage to using  $\beta_1>0$ instead of always $\beta_1 = 0$? The answer is yes. In App.C.1.8, we show that $\beta_1 = 0$ can be less stable. For this reason, we kept the two EMA formulation in our method, as there might be cases where $\beta_1>0$ is preferred. In general, we recommend using $\beta_1>0$ unless memory is an issue.

---

> > > ### Author Response · Authors · 2024-12-04
> > >
> > > Now answering other points raised by the reviewer:
> > >
> > > 1. Our Rosenbrock experiments only allow us to gain a bit of intuition by illustrating how (i) mixing slow and fast changing signals visibly results in less oscillations, (ii) using a single EMA with a large beta is not working, and (iii) AdEMAMix finds "good" solution relatively early. Beyond those observations, as mentioned before, the behavior on Rosenbrock is not necessarily an accurate representation of a high dimensional optimization landscape. The focus of this work is to develop a better optimizer for deep learning.
> > > 2. In line 249, we are talking about any exponential moving average over gradients. We are conveying that—in order to use an EMA with a very large beta—it is important to have another term which keeps its sensitivity to recent gradients.
> > >
> > > 3 & 5. We thank the reviewer for their interaction and providing suggestions on how to further improve our work.

---

### Meta-Review · Area_Chair_SnhG · 2024-12-20

**Metareview:**

This paper proposes a new heuristic to improve the momentum-based methods. In particular, it introduces another moving average sequence of stochastic gradient with a larger momentum parameter and adds it to the standard EMA sequence with a large weight. The paper has demonstrated superior performance in various settings. All reviewers agree that the paper has done a great work in demonstrating the effectiveness of the paper. However, a concern is that the paper does not provide any convergence guarantee of the proposed method. Hence, I will recommend a weak acceptance. The authors should take reviewers' comments into account for improving their paper.

**Additional Comments On Reviewer Discussion:**

The reviewers involved in the discussion with authors and acknowledged some concerns are addressed and also encouraged the authors to further improve the paper.

---

### Decision · Program_Chairs · 2025-01-22

Accept (Poster)